# NIa-Pro of sugarcane mosaic virus targets Corn Cysteine Protease 1 (CCP1) to undermine salicylic acid-mediated defense in maize

Wen Yuan[1☉], Xi Chen[1☉], Kaitong Du[1], Tong Jiang[1], Mengfei Li[1], Yanyong Cao[2], Xiangdong Li[3], Gunther Doehlemann[4], Zaifeng Fan[1], Tao Zhou[1]*

**1** State Key Laboratory for Maize Bio-breeding, and Ministry of Agriculture and Rural Affairs, Key Laboratory for Pest Monitoring and Green Management, Department of Plant Pathology, China Agricultural University, Beijing, China, **2** Cereal Crops Institute, Henan Academy of Agricultural Science, Zhengzhou, China, **3** Department of Plant Pathology, Shandong Agricultural University, Taian, China, **4** Institute for Plant Sciences and Cluster of Excellence on Plant Sciences (CEPLAS), University of Cologne, Center for Molecular Biosciences, Cologne, Germany

☉ These authors contributed equally to this work.
* taozhoucau@cau.edu.cn

**Data Availability Statement:** All relevant data are within the manuscript and its Supporting Information files.

## Abstract

Papain-like cysteine proteases (PLCPs) play pivotal roles in plant defense against pathogen invasions. While pathogens can secrete effectors to target and inhibit PLCP activities, the roles of PLCPs in plant-virus interactions and the mechanisms through which viruses neutralize PLCP activities remain largely uncharted. Here, we demonstrate that the expression and activity of a maize PLCP CCP1 (Corn Cysteine Protease), is upregulated following sugarcane mosaic virus (SCMV) infection. Transient silencing of CCP1 led to a reduction in PLCP activities, thereby promoting SCMV infection in maize. Furthermore, the knockdown of CCP1 resulted in diminished salicylic acid (SA) levels and suppressed expression of SA-responsive pathogenesis-related genes. This suggests that CCP1 plays a role in modulating the SA signaling pathway. Interestingly, NIa-Pro, the primary protease of SCMV, was found to interact with CCP1, subsequently inhibiting its protease activity. A specific motif within NIa-Pro termed the inhibitor motif was identified as essential for its interaction with CCP1 and the suppression of its activity. We have also discovered that the key amino acids responsible for the interaction between NIa-Pro and CCP1 are crucial for the virulence of SCMV. In conclusion, our findings offer compelling evidence that SCMV undermines maize defense mechanisms through the interaction of NIa-Pro with CCP1. Together, these findings shed a new light on the mechanism(s) controlling the arms races between virus and plant.

## Author summary

PLCPs perform crucial roles in plant defense against invasions by pathogens. While pathogens can secrete effectors that target and inhibit PLCP activities, the functions of PLCPs

**Funding:** This work was supported by National Natural Science Foundation of China (grants numbers 31371912 and 32072384 to T.Z.), Ministry of Agriculture and Rural Affairs of China (grants numbers NK2023070202, 2016ZX08010-001 and 2018YFD020062 to T.Z.), and Ministry of Education of China (the 111 Project B13006 to Z.F. and T.Z.). The funders had no role in study design, data collection and analysis, decision to publish, or preparation of the manuscript.

**Competing interests:** The authors have declared that no competing interests exist.

in plant-virus interactions and the mechanisms by which viruses counteract PLCP activities remain largely unexplored. In this study, we demonstrate that a maize PLCP termed CCP1 confers resistance to SCMV infection via SA signaling pathway. Conversely, SCMV NIa-Pro interacts with CCP1 and inhibits its protease activity to allow efficient viral infection. Furthermore, we have discovered that the critical amino acids responsible for the interaction between NIa-Pro and CCP1 are crucial for the virulence of SCMV.

## Introduction

Plants have evolved a sophisticated immune system to combat pathogen invasions [1,2]. Numerous studies have highlighted the critical roles papain-like cysteine proteases (PLCPs) play in regulating plant immunity against a range of pathogens, including bacteria, fungi, oomycetes, nematodes, and insects [3–9]. PLCPs, found ubiquitously in all living organisms, belong to the C1A family of peptidases cataloged in MEROPS, the Peptidase Database (https://merops.sanger.ac.uk/). The protease domain of PLCPs features a conserved catalytic triad composed of three amino acids (aa): Cys, His, and Asn. In plants, PLCPs are categorized into nine subfamilies based on their functional and structural attributes [10]. Typically, these pre-proteases exhibit an N-terminal signal peptide facilitating their entry into the endomembrane system and eventual activity in the apoplast, vacuoles, or lysosomes [9]. Under stress, PLCPs can act to induce a broad spectrum of responses, including plant cell death and salicylic acid (SA) signaling [11–15].

Conversely, pathogens have devised multiple stratagems to bypass plant immunity. A common tactic is the suppression of PLCP activities, often achieved through secretion of effectors or the deployment of endogenous inhibitors to stifle host defenses [4,5,12,16–21]. For example, the oomycete pathogen *Phytophthora infestans* has been shown to secret cystatin-like effectors (i.e., EPIC1 and EPIC2B) to inhibit the function of C14 protease in tomato and potato. Moreover, the C14 protease in tomatoes can be targeted by the *P. infestans* effector AvrBlb2, preventing its secretion into the apoplast and subsequent activation of a defense response [5]. In the fungal maize pathogen *Ustilago maydis*, the effector protein Pit2 impedes the induction of SA-mediated plant defense [20]. Pit2 operates as a substrate mimic, releasing an inhibitory peptide upon cleavage by apoplastic PLCPs [22]. In citrus plants, the Sec-delivered effector 1 (SDE1) from Huanglongbing-associated bacteria has been verified to inhibit the functions of immune-related PLCPs [23]. However, to date, our knowledge on how plant viruses regulate PLCP activities is still limited. Though a geminivirus tomato yellow leaf curl virus (TYLCV) was reported to interact with the tomato PLCP CYP1 and suppress its activity via its encoded V2 protein, the roles of PLCPs during viral infections in plants are still largely ambiguous [24, 25].

Plant viruses pose significant threats to global crop production. Given their compact genomes, viruses heavily rely on host factors and proteins adept at suppressing plant defenses to ensure successful life cycles. Potyviruses, belonging to the most expansive group of plant-infecting RNA viruses, have a widespread presence, infecting numerous crops, leading to considerable agricultural setbacks [26,27]. Potyviral genomes consist of single-stranded, positive-sense RNA that encodes a large polyprotein. Additionally, a P3N-PIPO, resulting from viral RNA polymerase slippage, is expressed within the P3 cistron [28,29]. This large polyprotein undergoes post-translational processing to yield ten mature proteins, including P1, HC-Pro, P3, and others [28,30,31].

As the main protease of the potyviruses, NIa is responsible for the processing and cleavage of all proteins except for P1 and HC-Pro [32,33]. Potyviral Nla is a protein that possesses two functional domains: an N-terminal VPg domain of 22 kDa and a C-terminal protease domain of 27 kDa (referred to as NIa-Pro) [32,33]. The VPg protein is typically covalently linked to the 5' end of the virus and facilitates the stability of the viral genome [33]. Additionally, the suboptimal cleavage between VPg and NIa-Pro is crucial to regulating their protein concentrations, and the slow release of VPg from the aggregated NIa protein is necessary for tobacco etch virus (TEV) infectivity [34]. VPg can regulate the structure and function of NIa-Pro, thereby adjusting the virus' enzyme activity to different stages of infection [35]. Although NIa-Pro possesses a cysteine residue in its active site, it shares structural motifs with eukaryotic cellular serine proteases [36]. Consequently, it is classified as a cysteine protease (MEROPS Clan PA, family C4) and is related to the 3C proteases of picornaviruses [37–40]. NIa-Pro is multifaceted, exhibiting non-sequence-specific DNase activity, binding to NIb for virus replication, and acting as a viral pathogenicity determinant [40,41]. It also enhances aphid vector growth and reproduction and has been identified to trigger the Ry-mediated disease resistance in potato [32,36,41–47]. A recent proteomic study revealed that the NIa-Pro of TEV interacts with 76 host proteins associated with plant stress responses, metabolism, and photosynthesis [48]. However, the precise mechanisms underlying NIa-Pro's function in virus infection remain elusive.

The potyvirus sugarcane mosaic virus (SCMV) is a prevalent viral pathogen that causes maize dwarf mosaic disease worldwide [49,50]. In China, SCMV infections have led to annual maize yield losses ranging from 10 to 50%, with the SCMV Beijing isolate being the dominant strain [51–53]. Notably, several maize proteins have been identified to directly interact with SCMV-encoded proteins [54–58]. To date, only three maize proteins (polyamine oxidase, violaxanthin de-epoxidase protein, and triacylglycerol lipase) have been characterized to confer resistance to SCMV infection [55,56,58]. There is a lack of reports on maize PLCPs' response to SCMV infection. Our recent study indicated that heightened SA accumulation upon infection bolsters maize's resistance against SCMV [59]. Given the pivotal role of PLCPs in directing SA-dependent defense signaling in maize [9], we postulated that SCMV might modulate maize PLCPs.

In this study, we demonstrated that SCMV infection in maize augments PLCP activity and specifically upregulates the expression of the PLCP gene, *Corn Cysteine Protease 1* (*CCP1*). We determined that *CCP1* imparts resistance against SCMV infection. Furthermore, we revealed that SCMV NIa-Pro interacts with CCP1, inhibiting its activity, which in turn diminishes CCP1's resistance and promotes SCMV infection. We also presented evidence highlighting the significance of two residues (K230 and D234) of NIa-Pro in its interaction with CCP1 and in SCMV's virulence.

## Results

### SCMV infection induces *CCP1* gene expression and activates PLCP activity

To ascertain whether PLCPs play a role in SCMV infection in maize, we initially probed the expression levels of PLCPs using our previous RNA sequencing dataset available in the Genome Sequence Archive (accession number CRA001815) from the Beijing Institute of Genomics Data Center (http://bigd.big.ac.cn/gsa) [60]. Our analysis revealed that the expression of the *CCP1* gene (*Corn Cysteine Protease 1*, *Zm00001d016446*) was significantly elevated at 3 and 9 days post SCMV inoculation (dpi). The *CCP2* gene (*Corn cysteine protease 2*, *Zm00001d020636*) also showed heightened expression at 3 dpi but not at 9 dpi. In contrast, SCMV infection did not influence the expression levels of other genes in the maize PLCP family (S1 Fig). Consequently, our attention was centered on *CCP1*.

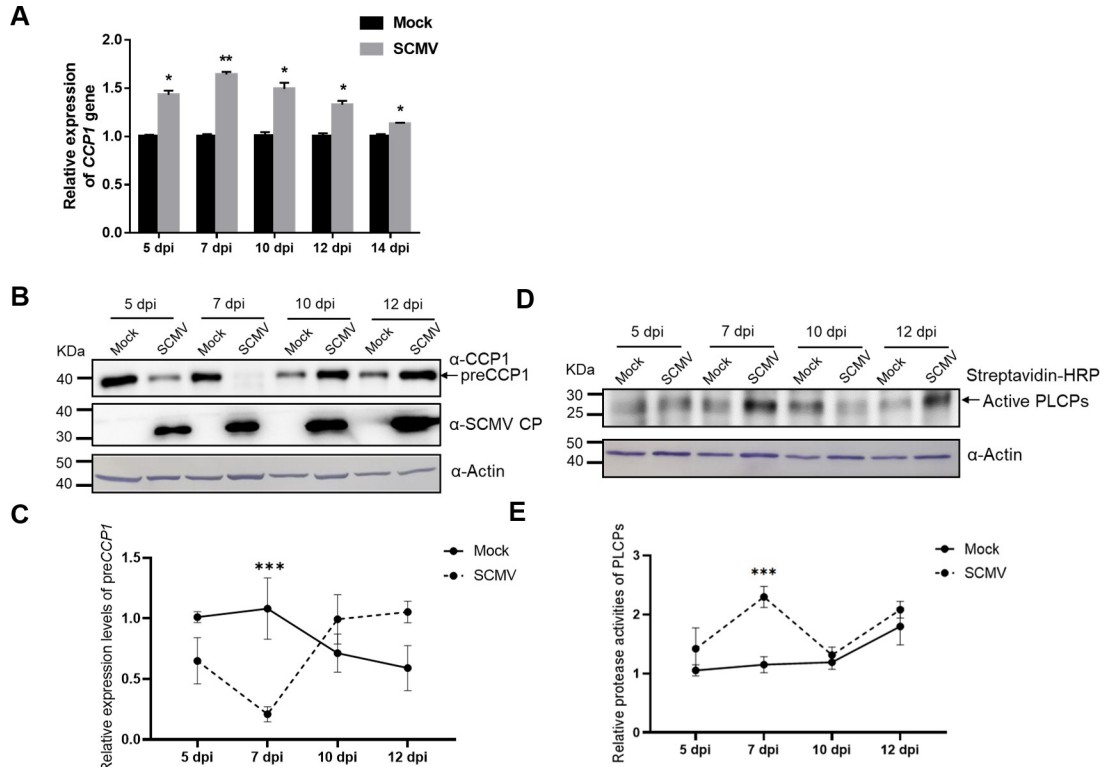

**Fig 1. SCMV infection up-regulates the expression of *CCP1* and alters the activities of maize PLCPs.** A) The expression of *CCP1* in maize was determined using quantitative reverse transcription PCR (RT-qPCR) at 5, 7, 10, 12 and 14 days post inoculation (dpi) of SCMV (gray bars). The plants inoculated with phosphate buffer (Mock, black bars) were used as controls. Statistical differences (*, $P < 0.05$; **, $P < 0.01$) were evaluated by Student's *t* test analysis. B) The accumulation of CCP1 precursor (preCCP1) in maize was determined by immunoblotting assay using antibody specific to CCP1 or SCMV CP at 5, 7, 10 and 12 dpi. C) The relative amount of preCCP1 was quantified by software ImageJ, normalized to actin control. And the lanes of 5 dpi Mock were set to 1.0. Data were presented as means ± SE (n = 4). Statistical differences (***, $P < 0.001$) were evaluated by two-tailed Student's *t* test analysis. D) The protease activities of maize PLCPs were determined by an activity-based protein profiling (ABPP). Only the active forms of PLCPs were labeled with DCG-04 and then detected using a streptavidin-conjugated with horseradish peroxidase (HRP). E) The relative accumulation of active PLCPs was quantified by software ImageJ, normalized to actin control. And the lanes of 5 dpi Mock were set to 1.0. Data were presented as means ± SE (n = 4). Statistical differences (***, $P < 0.001$) were evaluated by two-tailed Student's *t* test analysis. Three independent experiments are conducted.

The upregulation of *CCP1* in SCMV-infected maize plants was further validated using quantitative reverse transcription-PCR (RT-qPCR) at 5, 7, 10, 12, and 14 dpi (Fig 1A). Immunoblotting with a specific anti-CCP1 antibody indicated a downregulation of CCP1 precursor (preCCP1) during viral infection at 5 and 7 dpi, while an upregulation was observed at 10 and 12 dpi. This pattern suggests potential post-translational regulation of CCP1 in SCMV-infected maize plants.

Considering previous reports that PLCPs' proteolytic activity can sometimes diverge from their gene expression and protein abundance [18,23,61], we explored how SCMV infection influences PLCP protease activities. Using DCG-04, a biotinylated derivative of E-64 that binds exclusively to active forms of cysteine proteases, we conducted activity-based protein profiling (ABPP) [62,63]. The ABPP analyses revealed a temporary surge in PLCP activity at 7 dpi, which subsequently returned to the activity levels observed in mock-inoculated plants by 10 dpi (Fig 1D and 1E). Collectively, our findings demonstrate that SCMV infection alters maize PLCPs across gene expression, protein abundance, and protease activity.

### *CCP1* confers resistance to SCMV infection

To elucidate the role of *CCP1* in SCMV infection, we suppressed its expression in maize using virus-induced gene silencing (VIGS). We inserted a 210 bp fragment with high sequence specificity for *CCP1* into a cucumber mosaic virus (CMV, genus *Cucumovirus* in the family *Bromoviridae*)-based VIGS vector (S2 Fig) [64]. Maize seedlings treated with CMV-GUS served as negative controls. By 10 dpi, the plants treated with CMV-CCP1 exhibited no notable developmental aberrations. Subsequently, all the maize plants were subjected to challenge inoculation with SCMV. Relative to control plants, those with silenced *CCP1* manifested pronounced leaf chlorosis and a dwarfed growth post SCMV infection (Fig 2A). RT-qPCR data showcased that, in the first SCMV–systemically infected leaves, the expression level of *CCP1* in CMV-CCP1 silenced plants diminished by approximately 60% at 10 dpi when compared to SCMV-infected CMV-GUS control plants (Fig 2B). However, the level of SCMV genomic RNA in the

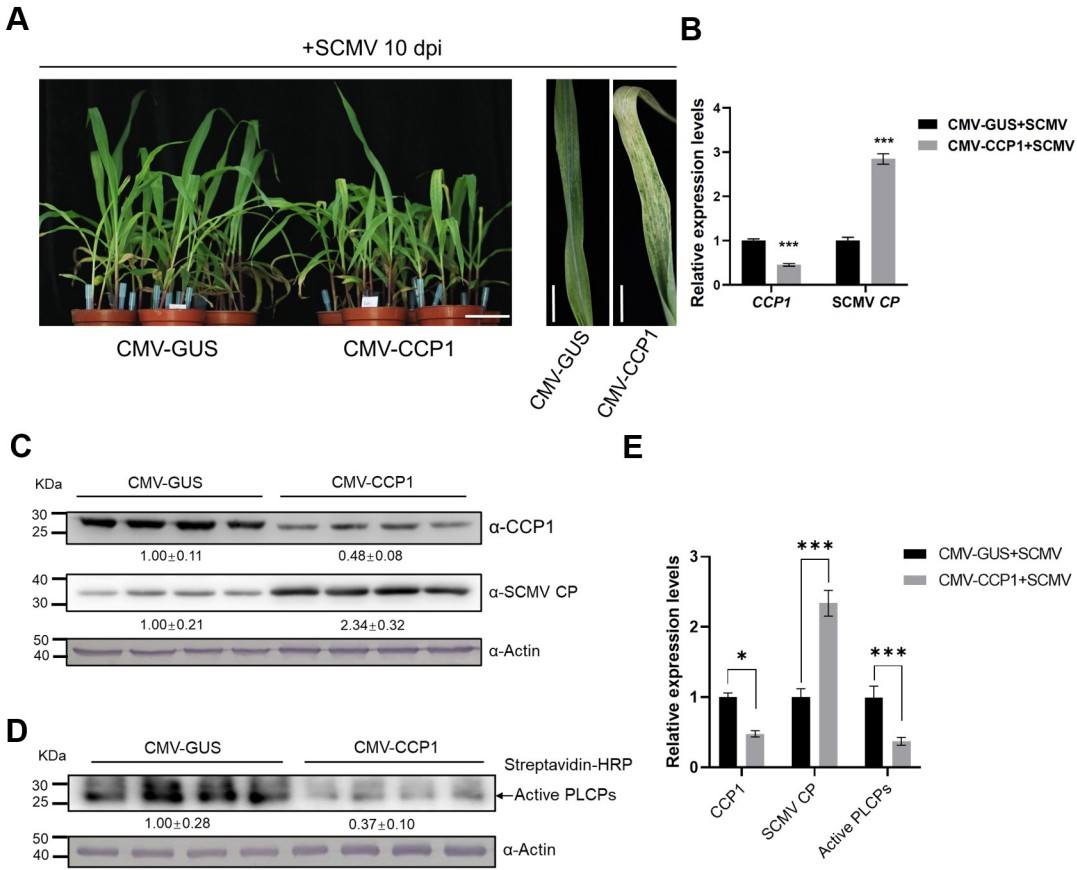

**Fig 2. Knockdown of *CCP1* in maize plants enhanced the accumulation level of SCMV.** A) The CCP1-silenced plants showed stronger leaf chlorosis and plant stunting compared with the CMV-GUS control plants at 10 d post challenge inoculation with SCMV. Bar = 5 cm. B) RT-qPCR analysis of the relative accumulation levels of *CCP1* and SCMV RNA in the SCMV first systemically infected leaves at 10 dpi. C) Immunoblotting analysis of CCP1 and SCMV CP accumulation levels. Actin was used as gel loading control. Band intensities were measured using software ImageJ. Numbers indicate the accumulation levels of CCP1 and SCMV CP normalized to actin control. D) Activity-based protein profiling (ABPP) analysis of the protease activities of maize PLCPs in the SCMV first systemically infected leaves at 10 dpi. Actin was used as gel loading control. Band intensities were measured using software ImageJ. Numbers indicate the accumulation levels of active PLCPs normalized to actin control. E) The relative accumulations of CCP1 (left), SCMV CP (middle) and active PLCPs (right) were quantified by software ImageJ, and the mean of CMV-GUS+SCMV was set to 1.0. Error bars represent the means ± SE. Statistical differences (*, $P < 0.05$; **, $P < 0.01$; ***, $P < 0.001$) were evaluated by two-tailed Student's *t* test analysis. Three independent experiments are conducted.

CCP1-silenced plants was augmented by about 1.7-fold compared to the controls (Fig 2B). The accumulation levels of CMV genomic RNA were similar in both groups (S4 Fig). Immunoblotting findings paralleled these results, revealing decreased CCP1 protein levels but heightened SCMV CP protein levels in the CCP1-silenced plants (Fig 2C and 2E). Moreover, ABPP demonstrated notably diminished PLCP protease activities in these plants (Fig 2D and 2E).

To further ascertain the role of *CCP1* in SCMV infection, we co-expressed CCP1 or a mutant CCP1 with three substitutions in its catalytic residues (termed *CCP1m*) and SCMV RNA in maize protoplasts. Protoplasts co-transfected with *mRFP* and SCMV RNA were used as a control. At 16 hours post transfection (hpt), we harvested the co-transfected protoplasts and subjected them to RT-qPCR analysis. Our analysis has revealed a 1.3-fold increase in the expression levels of CCP1 or CCP1m-overexpressed protoplasts (S5A Fig). Additionally, a substantial reduction of 0.5-fold in the accumulation of SCMV RNA was observed in the CCP1-overexpressed protoplasts, but not in the CCP1m-overexpressed protoplasts, compared to the protoplasts co-transfected with mRFP and SCMV RNA (S5B Fig). We have also examined the expression of SA marker genes Zm*PR1* and *ZmPR5* in protoplasts expressing CCP1, and found a significant increased for both (S5C and S5D Fig). Collectively, our evidence underscores the defensive role of CCP1 proteinase activity against SCMV infection in maize.

## CCP1 modulates SA signaling pathway

A recent report has shown that the PLCPs CP1A and CP2 can directly regulate SA-dependent signaling in maize [65]. Prompted by this, we sought to discern if CCP1 similarly modulates SA signaling by examining SA content and the transcription of SA-responsive pathogenesis-related genes (PRs) in both silenced and non-silenced maize. Upon *CCP1* suppression, no discernible growth alterations were observed relative to CMV-GUS controls (Fig 3A). Yet, SA content in *CCP1*-silenced plants was reduced by approximately 50%, while the levels of JA, ABA, and IAA remained consistent (Figs 3B and S3). Concurrently, the transcription levels of *ZmPR1* and *ZmPR5* in CCP1-silenced plants decreased by around 65% compared to controls (Fig 3C and 3D). These observations hint at CCP1's involvement in both SA accumulation and PR gene expression in maize.

In light of the up-regulation of *CCP1* expression by SCMV infection and the subsequent accumulation of SA, we further inspected the effect of SCMV infection on SA signaling in *CCP1*-silenced plants compared to CMV-GUS controls. Our findings revealed that, while SCMV infection augmented SA accumulation and *PRs* expression in both sets of plants, the levels in *CCP1*-silenced plants remained significantly lower than in controls (Fig 3B–3D). Since previous studies have demonstrated that SA can impede SCMV infection in maize [59], one can speculate that the suppression of SCMV infection by CCP1 is mediated by the modulation of SA signaling pathways.

## CCP1 interacts with SCMV NIa-Pro

To investigate if SCMV directly counteracts CCP1 to allow efficient infection, we employed yeast two-hybrid (Y2H) assays to identify SCMV protein(s) that might interact with CCP1. The CCP1 belongs to the subfamily 7 of PLCPs, which is homologous to the RD19 in *A. thaliana* [17]. Precursor CCP1 protein contains an N-terminal signal peptide (SP), an autoinhibitory pro-domain and the mature protease domain (Fig 4A). Firstly, we used the full-length CCP1 (preCCP1) to examine interactions with individual SCMV proteins, however, no interactions were observed in the transformed yeast cells. Considering that the protease domain of PLCPs was employed in the protein interaction assays in the maize-*U. Maydis* and citrus-Huanglongbing systems, we utilized the protease domain of CCP1 (aa residue 137–371,

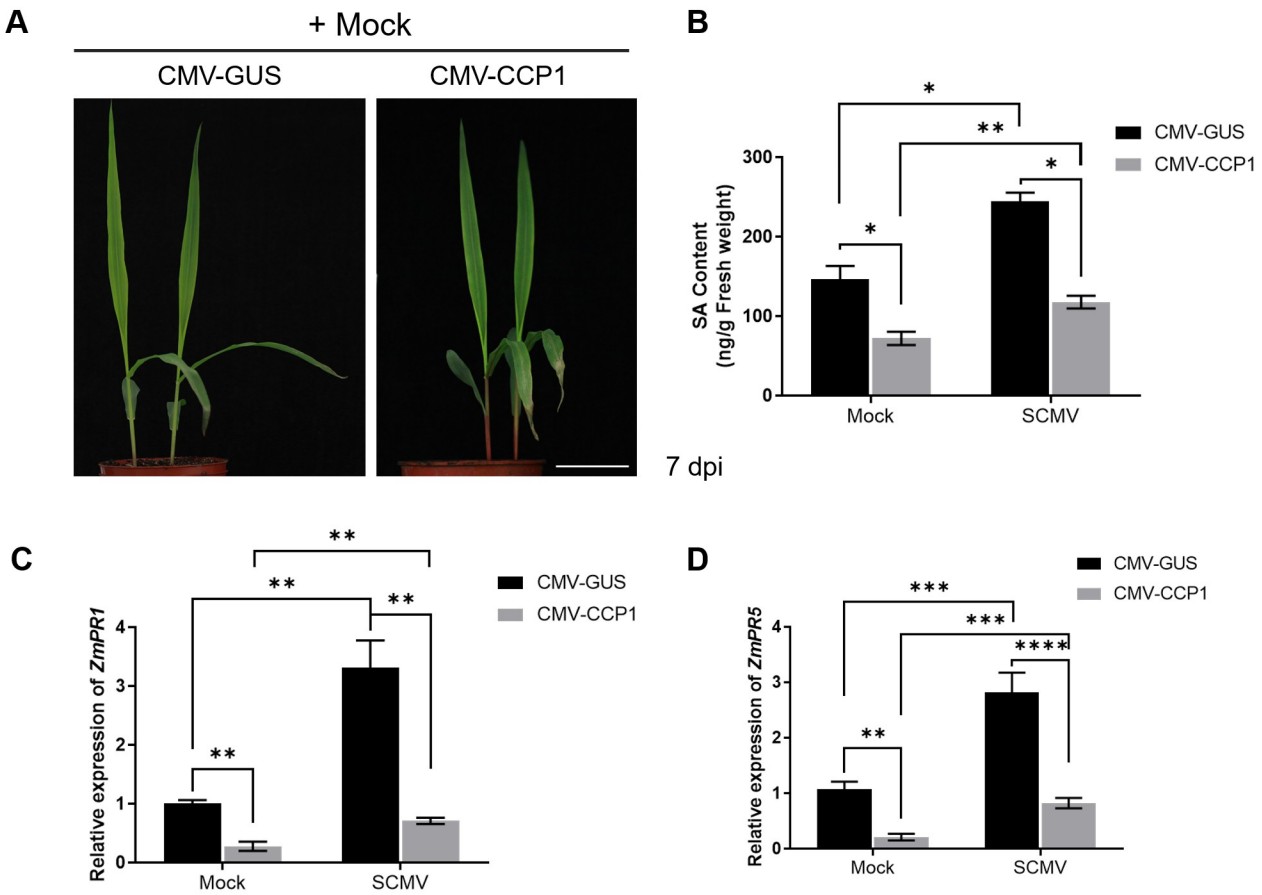

**Fig 3. Knockdown of *CCP1* in maize plants compromises SA biosynthesis and decreases *ZmPR1* and *ZmPR5* expressions.** A) Appearance of maize plants silenced for *CCP1* gene expression at 7 dpi. B) The contents of SA in the CMV-CCP1-inoculated or the CMV-GUS-inoculated maize plants at 7 d post challenge inoculation with SCMV or Mock. C) and D) Relative expression of *ZmPR1* (C) and *ZmPR5* (D) in the CMV-CCP1-inoculated or the CMV-GUS-inoculated maize plants. Three independent experiments were conducted with at least three biological replicates per treatment. Error bars represented the means ±SE. Significant differences between CMV-GUS and CMV-CCP1 infected plants are indicated (*, $P < 0.05$; **, $P < 0.01$; ***, $P < 0.001$; ****, $P < 0.0001$) were evaluated by Unpaired Student's *t* test analysis.

referred to as mCCP1) as a bait protein for Y2H to screen for interactions with each of the SCMV proteins [20,23]. Remarkably, we discovered that NIa-Pro is a potential interactor of mCCP1 (Fig 4B). On the other hand, neither the precursor of CCP1 (aa residue 1–371, preCCP1) nor the signal-peptide (SP) deletion mutant of CCP1 (aa residue 19–371, iCCP1) interacted with NIa-Pro in yeast cells (Figs 4A and S6). While there is no direct research evidence on the activation of CCP1, taking the example of a PLCP RD21 in *A. thaliana* [6,66], we speculate that CCP1 requires two steps to remove the signal peptide and auto-inhibitory domain, ultimately activating to its mature protease form.

Validating this interaction in *Nicotiana benthamiana* leaves, we confirmed that 3Flag-NIa-Pro interacts with both mCCP1-3Myc and preCCP1-3Myc (Figs 4C and S7). Leaves co-expressing mRFP-3Myc or GUS-3Myc and 3Flag-NIa-Pro served as negative controls (Figs 4C and S7). Furthermore, *in vivo* interaction between CCP1 and NIa-Pro was examined using a luciferase complementation assay (LCI). Co-expression of Nluc-mCCP1 or Nluc-preCCP1 and Cluc-NIa-Pro in *N. benthamiana* leaves led to a positive Luc activity (Figs 4D and S8). Moreover, *in vitro* pull-down assays provided evidence of a direct interaction between NIa-

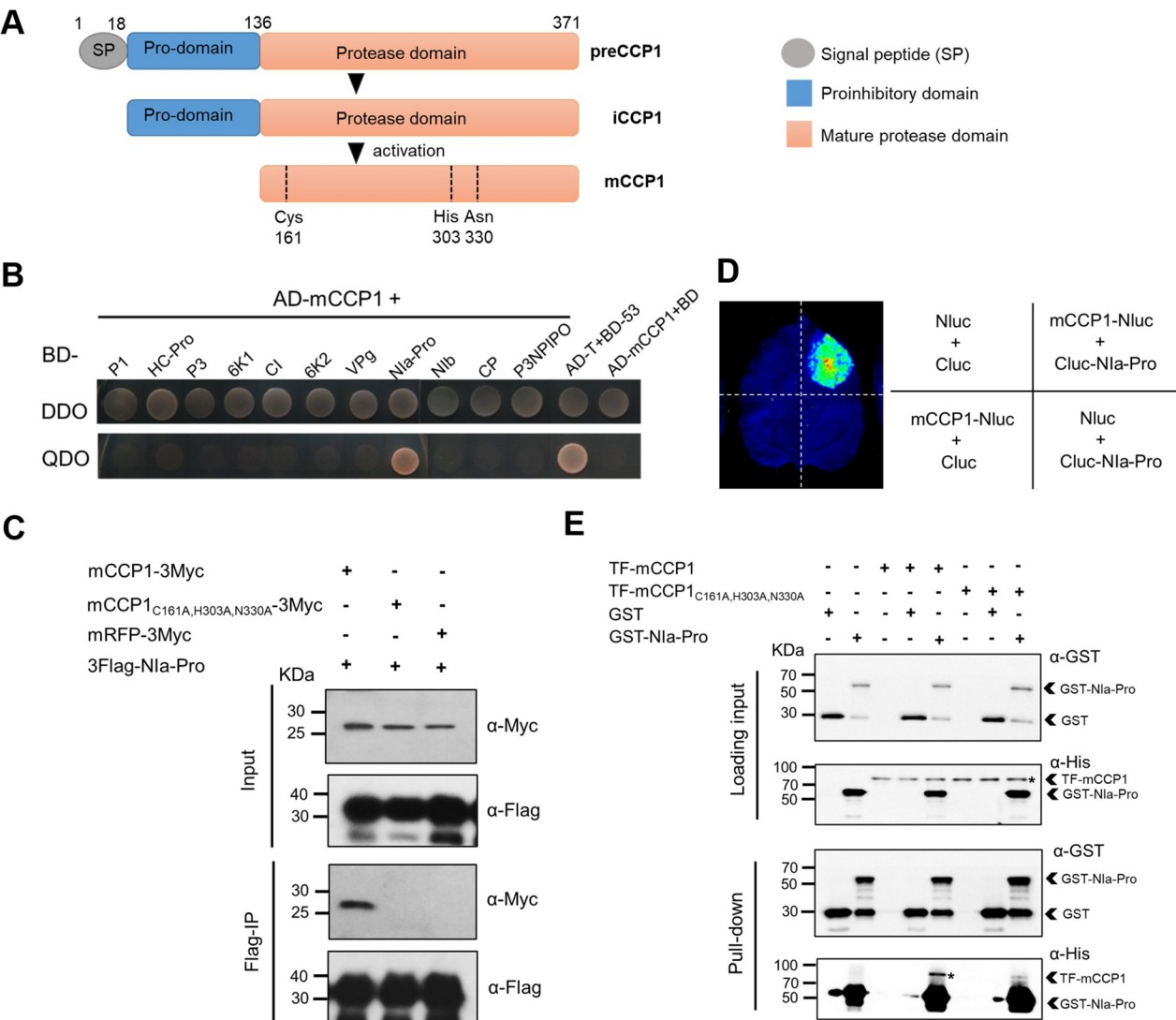

**Fig 4. SCMV NIa-Pro interacts with the mature protease domain of CCP1 (mCCP1).** A) A schematic diagram showing the domain structure of CCP1. CCP1 contains a signal peptide (SP), an autoinhibitory prodomain (pro-domain) and a mature protease domain. Position of the catalytic triad (Cys, His and Asn) in the mCCP1 is indicated. Different forms of CCP1 are named as preCCP1, iCCP1, and mCCP1, respectively. B) Pair-wise direct Y2H assay was conducted to examine the interaction between mCCP1 and the individual SCMV-encoded proteins. Yeast cells grown on the QDO selective medium are the cells with a positive protein–protein interaction. Yeast cells grown on the DDO medium are the transformed cells. Yeast cells transformed with the empty vector (BD) are used as a negative control. BD, Gal4 Binding Domain; AD, Gal4 activation domain. The AD-T+BD-53 serves as the positive interaction control. C) Co-immunoprecipitation (Co-IP) assay was conducted to verify the interaction between mCCP1 and NIa-Pro in *N. benthamiana* leaves. Mutant mCCP1$_{C161A,H303A,N330A}$ has three alanine substitutions at its catalytic triad. 3Flag-NIa-Pro was co-expressed with mCCP1-3Myc or mCCP1$_{C161A,H303A,N330A}$-3Myc in *N. benthamiana* leaves through agro-infiltration. Leaf tissues were harvested at 48 hours post agroinfiltration (hpai) for Co-IP analysis. Samples of plants co-expressing 3Flag-NIa-Pro and mRFP-3Myc were used as a negative control. Co-IP assay was performed using an anti-Flag affinity agarose gel. Protein samples (Input) and the immunoprecipitated protein samples (Flag-IP) were analyzed through immunoblotting assays using an anti-Flag or an anti-c-Myc antibody. D) Luciferase (Luc) complementation imaging (LCI) analysis of the interaction between NIa-Pro and mCCP1 in *N. benthamiana* leaves. The indicated plasmid pairs were transiently co-expressed in *N. benthamiana*. NLuc co-expressed with CLuc or CLuc-NIa -Pro, and mCCP1-NLuc co-expressed with CLuc, were used as negative controls. The luciferase activity was measured using a luminometer at 48 hpai. E) *In vitro* pull-down assay using a GST-tagged NIa-Pro to immunoprecipitate TF-tagged mCCP1 or mCCP1$_{C161A,H303A,N330A}$. GST-NIa-Pro was used as a bait in this assay and TF-mCCP1 or TF-mCCP1$_{C161A,H303A,N330A}$ was used as a prey. The input and the pull-down protein samples were analyzed through immunoblotting assays using an anti-GST or an anti-His antibody. Please note that the His Abs did show unspecific binding for the GST and GST-NIa-Pro proteins. *, the protein bands corresponding to the expected TF-mCCP1. Purified GST was used as a negative control.

Pro and mCCP1 (Fig 4E). These results confirm that NIa-Pro interacts with the protease domain of CCP1.

To better understand the mode of interaction, we asked whether NIa-Pro directly binds to the catalytic triad in CCP1. To this end, we generated a mutant mCCP1$_{C161A, H303A, N330A}$ where all residues of the catalytic triad were exchanged by Alanine. Strikingly, co-immunoprecipitation (Co-IP), GST Pull-down, Y2H, and LCI assays showed that mCCP1$_{C161A, H303A, N330A}$ did not interact with NIa-Pro (Figs 4C, 4E, S9 and S10). In addition, a fusion protein with VPg at the N terminus of NIa-Pro, known as NIa during SCMV infection in plant, also interacted with mCCP1 but not with mCCP1$_{C161A, H303A, N330A}$ (S11 Fig). These results suggest that the catalytic triad of CCP1 is critical for the interaction with NIa-Pro. To exclude the possibility of protein conformational changes caused by catalytic triad residues substitution of CCP1, we predicted the 3D-structure of mCCP1 and mCCP1$_{C161A, H303A, N330A}$ separately. The results suggest that the mutations have no effect on the protein conformation (S12 Fig).

To study the localization pattern, NIa-Pro was tagged at its C-terminus with GFP (NIa-Pro-GFP) and transiently expressed in *N. benthamiana* leaves through agroinfiltration. As expected, NIa-Pro was distributed in the cytoplasm and nucleus [46] (S13 Fig). Immunoblotting findings suggested that fusion protein NIa-Pro-GFP retains its integrity when compared to the control GFP (S14 Fig). Moreover, *A. thaliana* Aleurain (the marker for lytic vacuole) and HDEL (the marker for endoplasmic reticulum) were used for an elaborate investigation into the localization of CCP1. Similar to the observation of ortholog *A. thaliana* RD19, a perfect colocalization between CCP1 and Aleurain was discerned (S15A Fig) [67]. Also, a conspicuous colocalization of CCP1 with HDEL (S15B Fig) suggested that CCP1 is indeed situated on the endoplasmic reticulum initially, and may subsequently be directed to the lytic vacuole for activation.

We further co-expressed NIa-Pro-GFP and CCP1-mRFP for colocalization investigations in *N. benthamiana* and maize, revealing that partial colocalization within the cytoplasm of NIa-Pro and CCP1 occurs in the initial stages following agrobacterium infiltration (prior to 48 hours post-infiltration) or bombardment (S16A and S16C Fig), whilst in the latter phase of expression (from 48 to 72 hours post-infiltration), colocalization within the lytic vacuole can be observed in ~30% *N. benthamiana* cells (S16B Fig).

## NIa-Pro inhibits CCP1 protease activity

To elucidate the potential functional link between these two proteins, we investigated the impact of NIa-Pro on the stability of CCP1. The protein levels of preCCP1 or mCCP1 were assessed in *N. benthamiana* leaves co-expressing GUS and NIa-Pro, and no significant difference was observed in these samples (S17 Fig), indicating that NIa-Pro does not affect the stability of CCP1 *in vivo*. Given that NIa-Pro interacts with CCP1 through the protease domain, we asked whether it could inhibit CCP1 activity. Two assays were used to measure the proteolytic activities of CCP1 in the presence of NIa-Pro. In all these assays, a specific chemical inhibitor of PLCPs, E-64 was used as a positive control [62,68].

First, we examined the *in vitro* effect of NIa-Pro on CCP1 proteolytic activity using recombinant proteins purified from *E. coli*. Recombinant mCCP1 was expressed in *E. coli* with an N-terminal TF-tag, which was removed by thrombin cleavage and subsequent gel filtration. Concurrently, GST-tagged NIa-Pro or GST proteins were produced in *E. coli*. After pretreatment with purified GST, GST-NIa-Pro, or E-64, the relative activity of CCP1 was measured using a fluorescein-labeled casein substrate. When cleaved by PLCP, the casein substrate releases a fluorescent signal that can be quantified with a fluorometer. The results showed that the proteolytic activity of CCP1 was strongly inhibited by E-64 or GST-NIa-Pro pretreatment

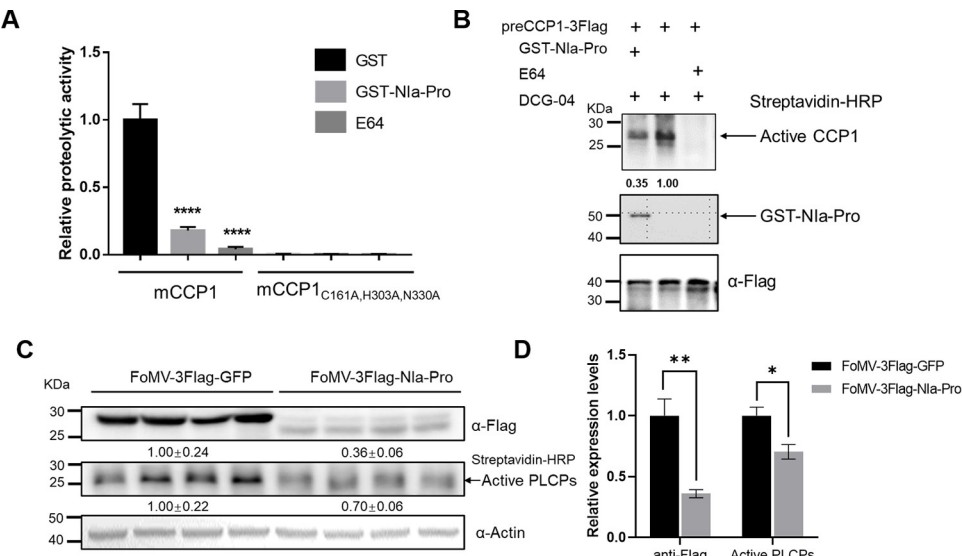

**Fig 5. NIa-Pro inhibits CCP1 activity *in vitro* and *vivo*.** A) Relative proteolytic activity of CCP1 was measured by digestion of a fluorescent casein substrate after pre-treatment with 5 μM E-64, 0.5 μM recombinant GST-NIa-Pro or GST (as negative control). In this assay, mutant mCCP1$_{C161A, H303A, N330A}$ with three alanine substitutions at its catalytic triad was used as an inactive control. Fluorescence at 485/530 nm (excitation/emission) was measured. The results are presented as means ± standard deviation (n = 3). Statistical differences between treatments were determined using the unpaired *t*-test. ****, $P < 0.0001$. B) ABPP was conducted to estimate the inhibitory effect of NIa-Pro on CCP1 protease activity. preCCP1-3Flag was transiently expressed in *N. benthamiana* leaves. Then the Flag-enriched proteins were incubated with 5 μM E-64 (as a positive control) or 1.0 μM purified GST-NIa-Pro followed by immunoblotting assay using a streptavidin-conjugated with horseradish peroxidase (HRP). ImageJ software was used to estimate the detection signal intensity. C) Expressing NIa-Pro via FoMV vector exhibited reduced PLCPs activity in maize. Active proteases were enriched using streptavidin beads and detected using streptavidin-HRP conjugates. Numbers indicate the accumulation levels of CCP1 and SCMV CP normalized to actin control. D) The relative accumulations of FoMV-overexpressing proteins (left), and active PLCPs (right) were quantified by software ImageJ, and the mean of FoMV-3Flag-GFP was set to 1.0. Error bars represent the means ± SE. Statistical differences (*, $P < 0.05$; **, $P < 0.01$) were evaluated by two-tailed Student's *t* test analysis. Three independent experiments are conducted.

compared to GST pretreatment (Fig 5A). The mutant mCCP1$_{C161A, H303A, N330A}$ exhibited no protease activity (Fig 5A), confirming the importance of the catalytic triad for CCP1 protease activity. When GST-NIa-Pro (0.5 μM) was added to the reaction, the proteolytic activity of CCP1 decreased by approximately 72% (Fig 5A). Its inhibitory effect was significant, though weaker than that of E-64, which reduced protease activity by 90%.

Next, we conducted ABPP in a semi-*in vitro* assay using recombinant NIa-Pro protein purified from *E. coli* and CCP1 expressed in plant tissues. The results demonstrated that CCP1 could be labeled by DCG-04, suggesting it is an active enzyme. As expected, a circa 0.65-fold reduction in CCP1 activity was observed with the addition of purified GST-NIa-Pro (1.0 μM) compared to the control (Fig 5B). The addition of E-64 completely abolished the DCG-04 labeling.

We then ascertained whether NIa-Pro could inhibit PLCPs protease activities in maize. A heterologous foxtail mosaic virus (FoMV, genus *Potexvirus* in the family *Alphaflexiviridae*) vector was used for NIa-Pro expression [69]. Coding sequences of full-length SCMV NIa-Pro fused to 3Flag were inserted into a FoMV-based expression vector to produce FoMV-3Flag-NIa-Pro, with FoMV-3Flag-GFP set as a control. Total proteins were extracted from FoMV-infected maize leaves for analysis. The results showed that, though the accumulation level of 3Flag-NIa-Pro was significantly lower than that of 3Flag-GFP, PLCP activity was markedly

reduced in NIa-Pro-expressed (FoMV-3Flag-NIa-Pro) plants compared to the control (FoMV-3Flag-GFP) plants (Fig 5C and 5D). Collectively, NIa-Pro plays a role in suppressing the protease activity of PLCPs.

## The 48-amino acid inhibitor motif is sufficient for function of NIa-Pro to interact with CCP1 and suppress its activity

To pinpoint the essential amino acid residues responsible for suppressing CCP1 activity, we predicted the structure of SCMV NIa-Pro using its complete amino acid sequence in the RCSB PDB (http://www.rcsb.org/). The analysis revealed that NIa-Pro contains an N-terminal protease domain (pdNIa-Pro, 1–194 aa) (Fig 6A and 6B). As observed in crystal structures of several potyviral protease, the C-terminus of SCMV NIa-Pro (195–242 aa) is disordered (Fig 6B) [70, 71]. Previous studies have discovered that the C-terminus of NIa-Pro encoded by TEV, genus *Potyvirus* binds to its active site [70]. The full-length form of NIa-Pro encoded by TVMV (tobacco vein mottling virus, genus *Potyvirus*) exhibits significantly lower activity than its C-terminus-deleted form, suggesting that the C-terminus of NIa-Pro might potentially exert inhibitory effects on its activity. The predicted structure of SCMV NIa-Pro also suggested that the speculated inhibitory motif is surface-localized, potentially facilitating its association with CCP1.

Subsequently, we divided NIa-Pro into two regions based on its predicted structure: the protease domain (pdNIa-Pro, 1–194 aa) and the speculated inhibitory motif (imNIa-Pro, 195–242 aa) (Fig 6A). An LCI assay confirmed that imNIa-Pro (195–242aa) alone could interact with mCCP1 (Fig 6C). Further analysis of the amino acid sequences of NIa-Pro from various potyviruses showed that the speculated inhibitory motif is present in other potyviruses, with several residues being highly conserved across multiple NIa-Pros (S18 Fig). While W199 and W203 are strictly conserved in the aligned *Potyvirus* species, residues N206, K230, and D234, although not strictly conserved, are conserved in the majority of potyviruses. To identify the specific amino acid(s) responsible for the NIa-Pro–CCP1 interaction, we introduced single point mutations (W199A, W203A, N206A, K230A, and D234A) into the inhibitory motif. The LCI assay and Y2H results revealed that residues K230 and D234 are both vital for the NIa-Pro–CCP1 interaction (Figs 6D and S19).

These findings suggest that the inhibitor motif alone is sufficient for the NIa-Pro–CCP1 interaction, with residues K230 and D234 being essential. We hypothesized that the inhibitor motif alone could interfere with CCP1 activity. To test this, we used purified recombinant protein GST-imNIa-Pro and GST-NIa-Pro$_{K230A, D234A}$ for protease activity experiments. The results showed that GST-imNIa-Pro reduced the proteolytic activity of CCP1 by approximately 80%, a more significant reduction than the full-length NIa-Pro (Fig 6E). The mutated NIa-Pro$_{K230A, D234A}$ did not significantly affect CCP1 activity (Fig 6E).

## The essential role of NIa-Pro's key residues in SCMV infection

To elucidate the significance of the inhibitor motif and the key residues of NIa-Pro in SCMV infection, we incorporated the mutations K230A and D234A into the NIa-Pro of the SCMV-GFP infectious clone, resulting in the variant pSCMV-NIa-Pro$_{K230A, D234A}$-GFP. Concurrently, we excised the inhibitor motif of NIa-Pro from the SCMV-GFP infectious clone, yielding the clone pSCMV-NIa-Pro$_{IMD}$-GFP. Agrobacterium cells harboring pSCMV-NIa-Pro$_{K230A, D234A}$-GFP, pSCMV-NIa-Pro$_{IMD}$-GFP, and pSCMV-GFP were then infiltrated into *N. benthamiana* leaves. By 5 days post-agroinfiltration (dpai), GFP fluorescence was discernible in areas infected with SCMV-GFP or SCMV-NIa-Pro$_{K230A, D234A}$-GFP (Fig 7A). Notably, the GFP fluorescence intensity in areas infected with SCMV-NIa-Pro$_{IMD}$-GFP was markedly

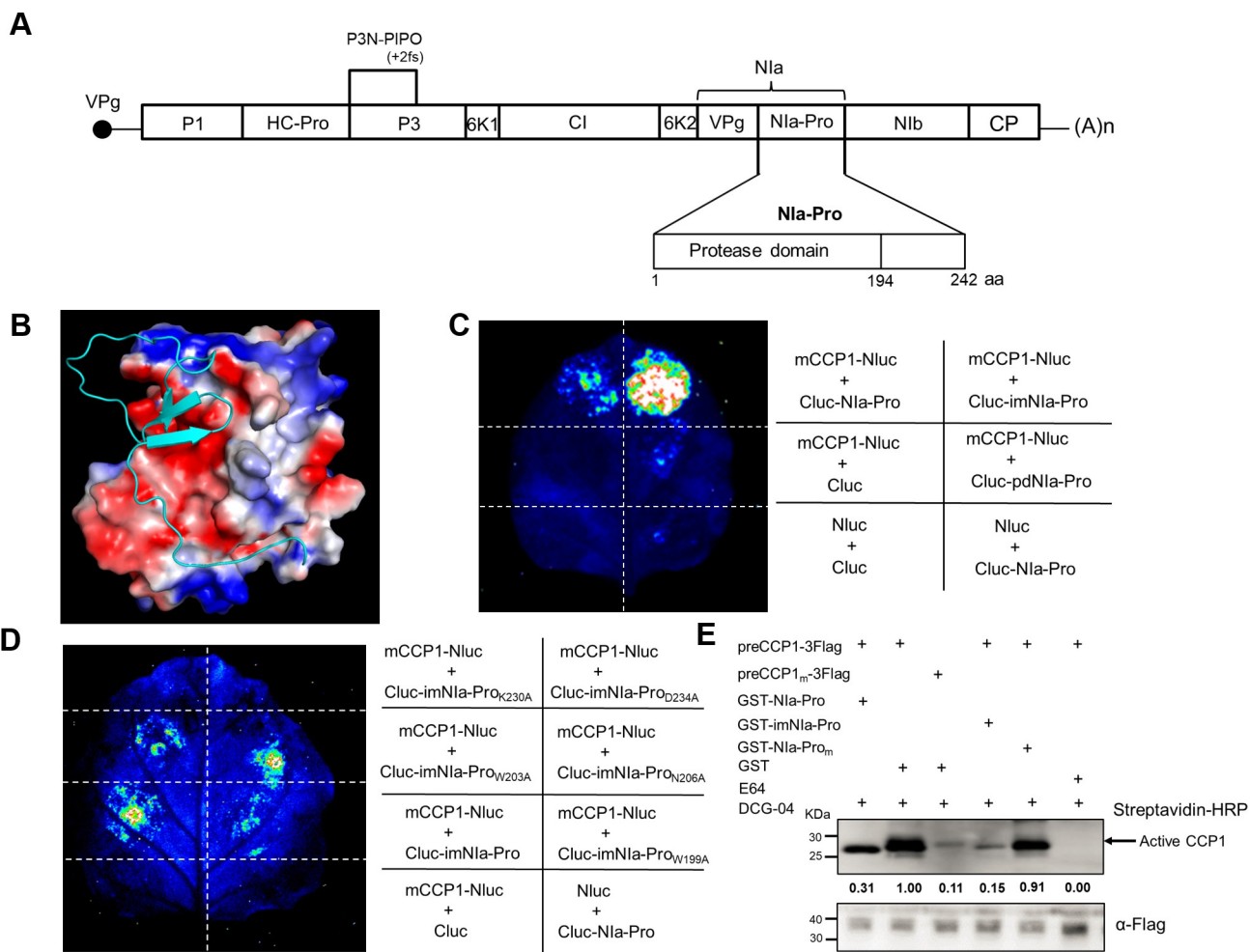

**Fig 6. Identification of the key amino acids of NIa-Pro for interaction with CCP1.** A) Schematic representation of NIa-Pro in SCMV genome. SCMV protein: P1: the first protein; HC-Pro: Helper Component-proteinase; P3: the third protein; 6K1: the first 6K protein; CI: cytoplasmic inclusion protein, 6K2: the second 6K protein, VPg: viral genome-linked protein, NIa-Pro: nuclear inclusion protein a-proteinase, NIb: nuclear inclusion protein b, CP: coat protein, P3N-PIPO: the N-terminal half of P3 fused to Pretty Interesting Potyvirus Open Reading Frame. B) The predicted 3D structure of SCMV NIa-Pro shows a protease domain (Cartoon shape) and an inhibitor motif (the blue line indicated). C) LCI assay of the interaction between mCCP1 and pdNIa-Pro or imNIa-Pro in *N. benthamiana* leaves. The Agrobacterium strains carrying the indicated constructs were infiltrated into *N. benthamiana* leaves. D) LCI assay for identification of the amino acid within the imNIa-Pro essential for interaction with mCCP1. The Agrobacterium strains carrying the indicated constructs were infiltrated into *N. benthamiana* leaves. NLuc co-expressed with CLuc or CLuc-NIa-Pro-, and mCCP1-NLuc co-expressed with CLuc, were used as negative controls. The luciferase activity was measured using a luminometer at 48 hpai. E) ABPP analysis to estimate the inhibitory effect of different form NIa-Pro on CCP1 protease activity. preCCP1-3Flag or preCCP1$_{C161A,H303A,N330A}$ (preCCP1$_m$)-3Flag was transiently expressed in *N. benthamiana* leaves. Then the Flag-enriched proteins were incubated with 5 μM E-64 (as a positive control) or 1.0 μM purified GST-NIa-Pro, GST-NIa-Pro$_{K230A, D234A}$(NIa-Pro$_m$), GST-imNIa-Pro and GST (as a negative control), followed by immunoblotting assay using a streptavidin-conjugated with horseradish peroxidase (HRP). PreCCP1$_m$-3Flag was used as an inactive control. ImageJ software was used to estimate the detection signal intensity.

diminished compared to regions infected with SCMV-GFP and SCMV-NIa-Pro$_{K230A, D234A}$-GFP (Fig 7A). RT-qPCR analyses further corroborated that the relative accumulation of SCMV mRNA in areas infected with SCMV-NIa-Pro$_{IMD}$-GFP was substantially reduced compared to those infected with SCMV-GFP or SCMV-NIa-Pro$_{K230A, D234A}$-GFP (Fig 7B).

Subsequent inoculation of maize seedlings with crude extracts from SCMV-infected *N. benthamiana* leaves revealed pronounced mosaic symptoms in maize plants infected with SCMV-GFP by 8 dpi. In contrast, the virulence of the two mutant SCMV on maize was

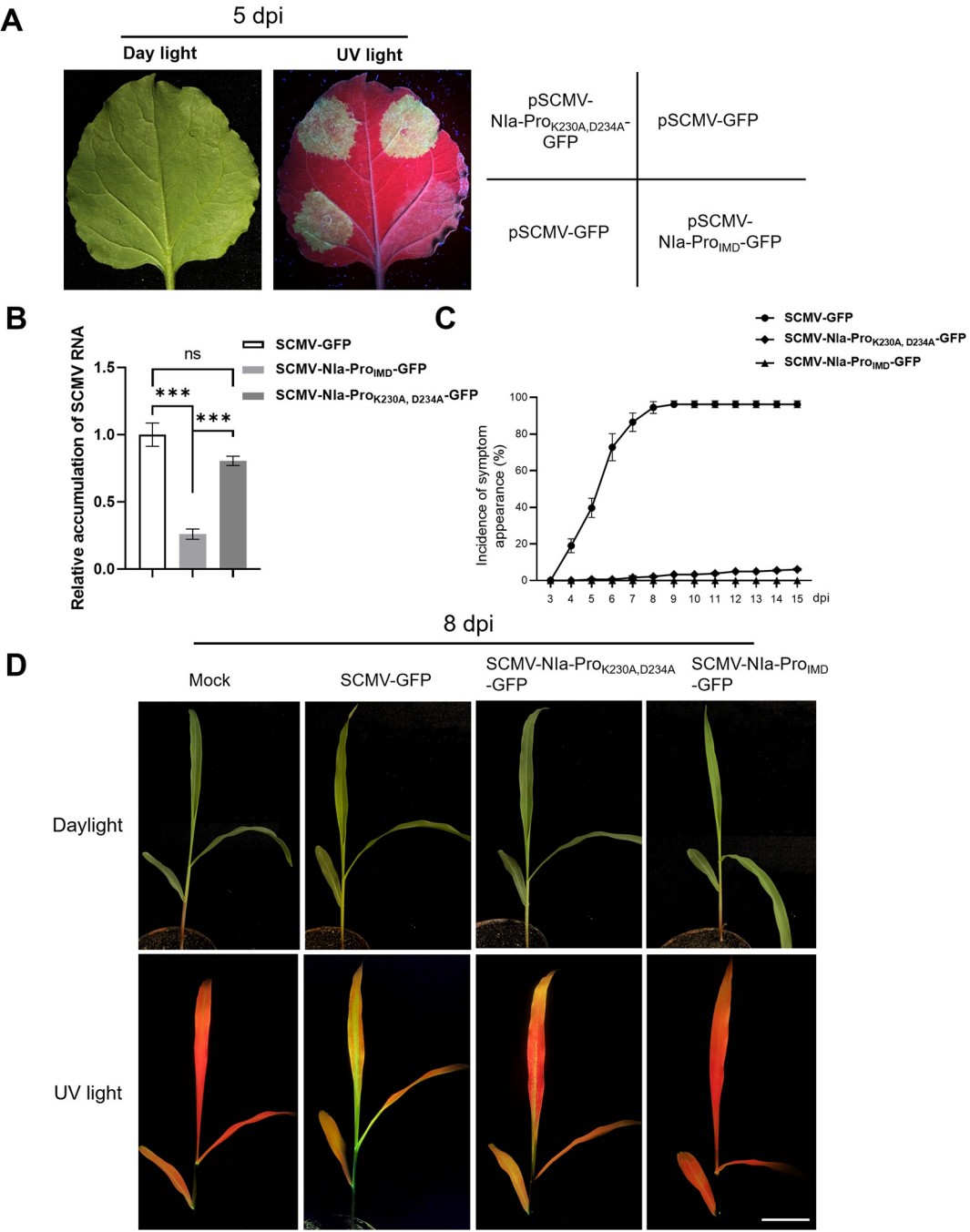

**Fig 7. Truncated NIa-Pro or key amino acids mutant NIa-Pro$_{K230A, D234A}$ inhibited SCMV infection in maize plants.** A) GFP analysis of the *N. benthamiana* leaves agro-infiltrated with the wild type virus SCMV-GFP, the inhibitor motif of NIa-Pro-defective mutant SCMV-NIa-Pro$_{IMD}$-GFP, and substitutions mutant SCMV-NIa-Pro$_{K230A, D234A}$-GFP at 5 dpai. Under UV light, strong GFP fluorescence was evident in pSCMV-GFP and pSCMV-NIa-Pro$_{K230A,D234A}$-GFP infected areas. No GFP fluorescence was observed on pSCMV-NIa-Pro$_{IMD}$-GFP infected areas. B) RT-qPCR analysis of the relative accumulation levels of SCMV RNA in the SCMV agro-infiltrated leaves at 5 dpi. Error bars represent the means ± SE. Statistical differences (**, $P < 0.01$) were evaluated by Student's *t* test analysis. Three independent experiments are conducted. C) Statistical analysis of the incidence of symptoms appearance on maize plants. D) Symptoms of maize plants infected with SCMV-GFP, and its NIa-Pro mutants SCMV-NIa-Pro$_{IMD}$-GFP or SCMV-NIa-Pro$_{K230A, D234A}$-GFP at 8 dpi. Bar = 5 cm.

significantly impacted. The SCMV-NIa-Pro$_{IMD}$-GFP did not show any symptoms even at 15 dpi, whereas the double mutant virus SCMV-NIa-Pro$_{K230A, D234A}$-GFP exhibited a disease incidence of less than 4% at 15 dpi (Fig 7C). The GFP fluorescence and mosaic symptoms on systemic leaves of SCMV-NIa-Pro$_{K230A, D234A}$-GFP infected maize plants were noticeably weaker compared to SCMV-GFP (Fig 7D). Presence of intact viral proteins in the mutant SCMV-NIa-Pro$_{K230A, D234A}$-GFP-infected maize plants suggested that the NIa-Pro proteinase activity of SCMV-NIa-Pro$^{K230A, D234A}$-GFP is unimpaired (S20 Fig). All these results suggested the essential role of NIa-Pro's key residues in SCMV virulence.

### The double mutant virus SCMV-NIa-Pro$_{K230A, D234A}$-GFP fails to induce *CCP1* gene expression or activate PLCP activity

Due to successful infection of only the double mutant virus SCMV-NIa-Pro$_{K230A, D234A}$-GFP in a small number of maize plants (4%), we further compared the virus accumulation and the corresponding gene expression levels in the mutant virus and original virus-infected maize plants. The RT-qPCR results indicated that the accumulation of SCMV RNA in maize plants infected with SCMV-NIa-Pro$_{K230A, D234A}$-GFP decreased to approximately 0.2 times that of the original virus SCMV-GFP (Fig 8A). Additionally, immunoblotting analyses further confirmed a significant decrease of CP accumulation levels of SCMV-NIa-Pro$_{K230A, D234A}$-GFP compared with SCMV-GFP-infected maize plants (Fig 8B). These observations underscore the pivotal role of residues K230 and D234, which mediate the NIa-Pro–CCP1 interaction, in facilitating SCMV infection in maize.

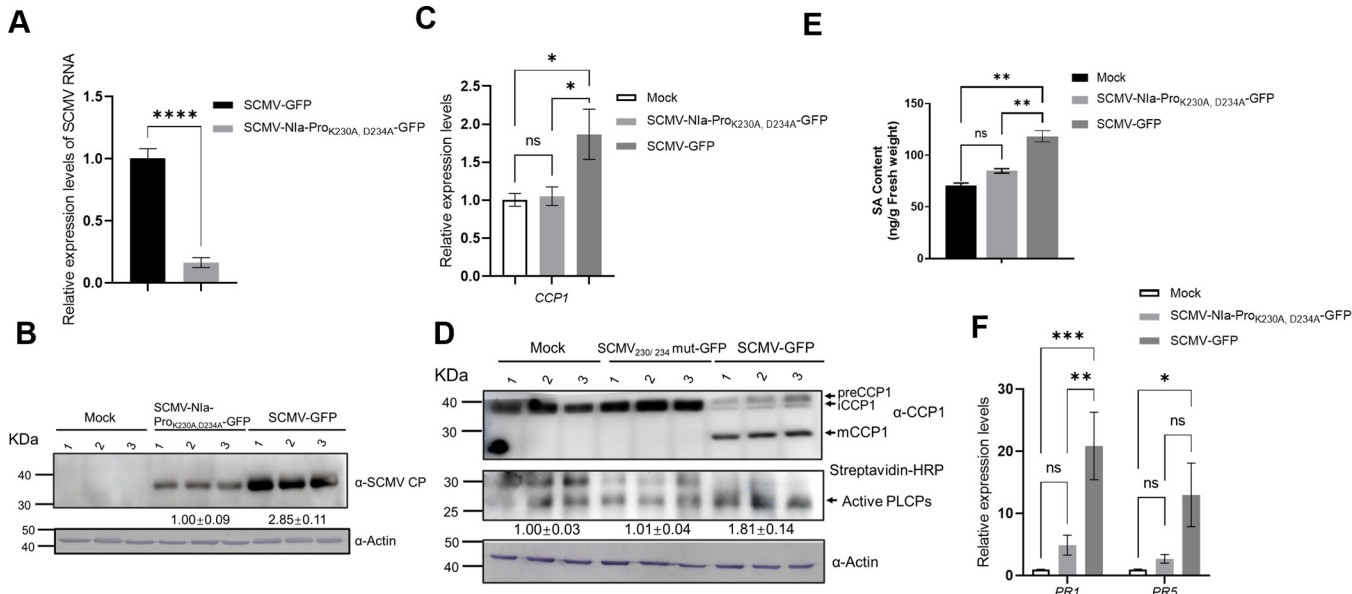

**Fig 8. The double mutant virus SCMV-NIa-Pro$_{K230A, D234A}$-GFP fails to induce CCP1 gene expression and activates PLCP activity.** A) RT-qPCR analysis of the relative accumulation levels of SCMV RNA in the SCMV-GFP- or SCMV-NIa-Pro$_{K230A, D234A}$-GFP- infected first systemically leaves at 8 dpi. B) Immunoblotting analysis of SCMV CP accumulation in the first systemically infected maize leaves. C) RT-qPCR analysis of the relative accumulation levels of *CCP1* in the mock, SCMV-GFP- or SCMV-NIa-Pro$_{K230A, D234A}$-GFP- infected first systemically leaves. D) Immunoblotting analysis of CCP1 accumulation and activity-based protein profiling (ABPP) analysis of the protease activities of maize PLCPs in the first systemically infected leaves at 8 dpi. E) The contents of SA in the mock, SCMV-GFP- or SCMV-NIa-Pro$_{K230A, D234A}$-GFP- infected first systemically leaves. F) RT-qPCR analysis of the relative accumulation levels of *PR1* and *PR5* in the mock, SCMV-GFP- or SCMV-NIa-Pro$_{K230A, D234A}$-GFP- infected first systemically leaves. Error bars represent the means ± SE. Statistical differences (*, $P < 0.05$; **, $P < 0.01$; ***, $P < 0.001$; ****, $P < 0.0001$) were evaluated by two-tailed Student's $t$ test analysis. Three independent experiments are conducted.

At the same time, we measured the levels of CCP1 accumulation and PLCPs enzyme activity in maize plants. The results showed that, compared with the significant upregulation of *CCP1* transcriptional levels in maize plants infected with SCMV-GFP, there were no significant changes in *CCP1* expression levels in maize plants infected with the SCMV-NIa-Pro$_{K230A, D234A}$-GFP, which were comparable to those in the mock group (Fig 8C). Additionally, we detected the accumulation levels of active CCP1 and found that CCP1 was significantly activated in maize infected with SCMV-GFP, whereas there was no significant induction observed in the SCMV-NIa-Pro$_{K230A, D234A}$-GFP infected group (Fig 8D).

Analysis of active PLCPs also revealed a significant increase in maize plants infected with SCMV-GFP at 8 dpi, whereas there was no such increase in the SCMV-NIa-Pro$_{K230A, D234A}$-GFP -infected group (Fig 8D). Lastly, measurement of SA hormone levels and the expression of SA marker genes showed that there was no significant induction in SA levels or the upregulation of *PR1* and *PR5* in the SCMV-NIa-Pro$_{K230A, D234A}$-GFP-infected group, despite the increase observed in the SCMV-GFP-infected group (Fig 8E and 8F).

## Discussion

In this study, we found that SCMV infection enhances CCP1 activation and induces SA signaling pathway activation in maize plants. The activated CCP1 mediates maize resistance to SCMV and is associated with SA signaling pathway, whereas the precise regulatory mechanism remains unknown. Conversely, SCMV-encoded NIa-Pro interacts with CCP1 to inhibit its activity, thereby weakening resistance to SCMV. Thus, we proposed a working model to illustrate these findings (Fig 9).

PLCPs are known to play important roles in regulating plant defense against a broad range of microbial pathogens including bacteria, fungi and oomycetes [7,23,66,72–74]. In this study, we have demonstrated the upregulation of CCP1 in maize upon SCMV infection, and this upregulation contributes to the resistance against SCMV infection. We found that the expression of *CCP1* is up-regulated upon SCMV infection in maize leaves. In line with this, PLCP activity was induced early after SCMV infection. This is supported by the analyses of published transcriptome data from several virus-infected plants, which revealed that the expression of PLCPs can be up-regulated upon virus infection in various species (S1 Table) [75–80]. Through knock-down assays using CMV-based VIGS vector, we have shown that CCP1 plays a role in maize resistance to SCMV infection. Given the resistance roles of PLCPs in fungal or bacterial infected plants, it is suggested that the PLCPs-mediated resistance is conserved for against diverse biotic stresses [9,16,81].

The roles of plant proteases in disease resistance have been documented, yet the mechanisms underpinning these resistances remain elusive [18,82]. Some studies suggest that PLCPs might directly hydrolyze pathogen components. For instance, papain was found to curb the growth of *Phytophthora palmivora* in papaya [83]. Additionally, PLCPs have been noted to cleave plant peptides, initiating host defense responses [9]. In maize, SA prompts the activation of PLCPs, which subsequently triggers SA-dependent defense mechanisms [12]. The activation of SA-induced PLCPs results in the release of Zip1, a bioactive immune signaling peptide derived from its propeptide precursor. This peptide acts as an activator of SA biosynthesis [65]. Notably, the Zip1-mediated activation of SA signaling necessitates the apoplastic proteases CP1A, a member of the PLCP Subfamily 1, and CP2 from Subfamily 8. In related research, increased activities of PIP1 and RCR3 in tomatoes post benzothiadiazole treatment, an SA analog [18]. The potential of other PLCPs to modulate SA signaling is yet to be ascertained. Our findings show that silencing CCP1 expression in maize markedly reduces the levels of endogenous SA and the expression of SA-responsive genes *ZmPR1* and *ZmPR5*. Given that SA

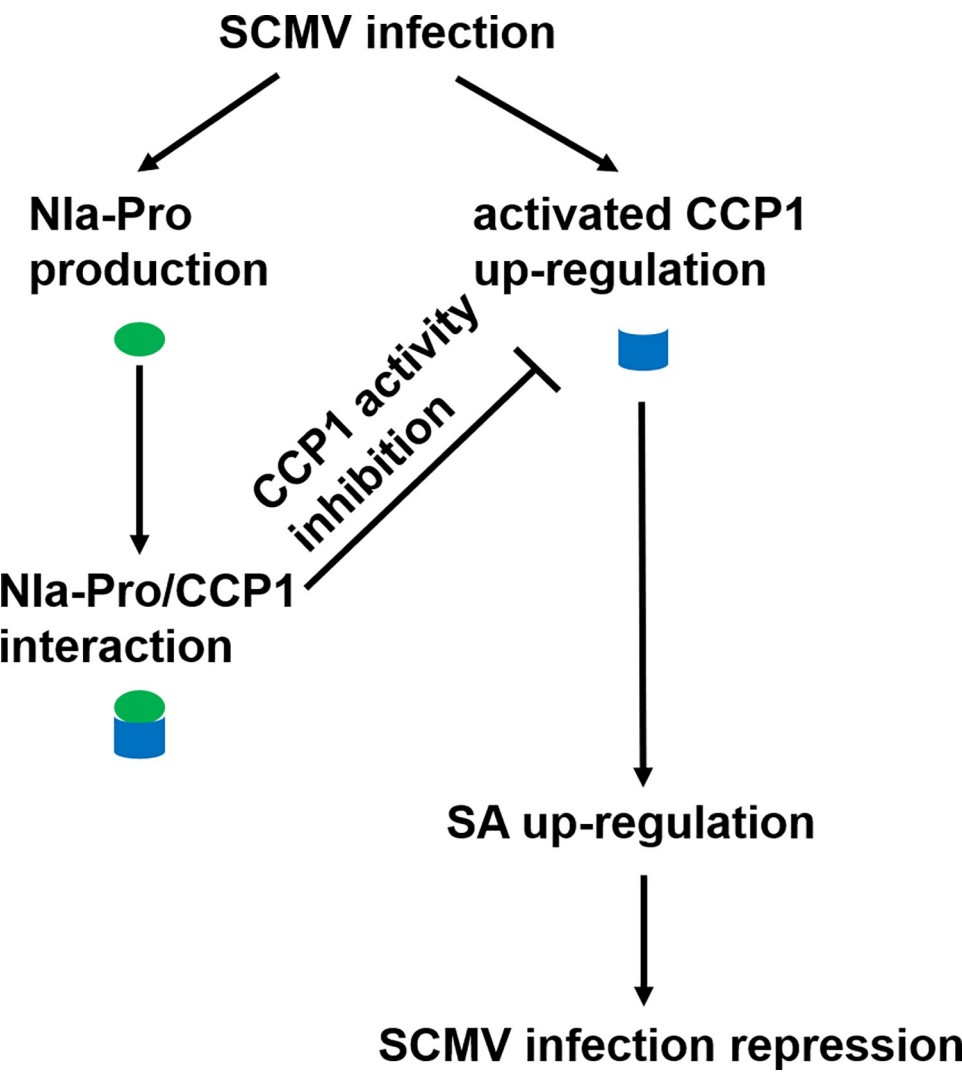

**Fig 9. Proposed model of NIa-Pro targeting Corn Cysteine Protease 1 (CCP1) for efficient infection of SCMV in maize.** CCP1 is a maize PLCP which confers resistance to SCMV infection via activation of SA signaling pathway. NIa-Pro is the main viral protease of SCMV. NIa-Pro can interact with CCP1 and inhibit its activity to counteract CCP1-mediated resistance and facilitate SCMV infection in maize plants.

content rises in SCMV-infected maize plants and that activated SA signaling plays a crucial role in maize's resistance against SCMV infection [59], we postulate that CCP1-mediated resistance to SCMV is moderated via SA signaling regulation in maize.

Our findings in this study suggest that CCP1 contributes to SA-associated defense in maize, which in turn is important for plant resistance to SCMV. However, it still remains to be answered how CCP1 interferes with SCMV infection on the cellular level. Besides the induction of SA-signaling, PLCPs have also been reported to be involved in the induction of plant cell death responding to pathogen infection and leaf senescence [11, 13, 14]. Silencing CCP1 does not activate PLCP-mediated SA signaling, thus the virus replicates in higher level and causes more severe symptom.

For infection to be compatible, plant PLCPs need to be inhibited by pathogen effectors [20]. Ample evidence reveals that various pathogens, including bacteria, fungi, oomycetes, and nematodes, have evolved to produce effectors that hinder PLCP activities, thus aiding their

infection in plants [5,7,16,23,66,72,74,84]. While PLCPs are targeted by specific effectors, the inhibiting effectors do not possess conserved motifs, indicating that these effectors have evolved independently to disrupt the activity of particular PLCPs [23]. In our investigation, we found that the localization pattern of CCP1 is similar to that of vacuolar proteases that are synthesized as preproteins in the rough endoplasmic reticulum and transiently transported to the vacuole through the endomembrane secretion system [85], ultimately exerting its protease activity in the vacuole. In addition, we found that SCMV NIa-Pro constrains CCP1 protease activity through its interaction with the active protease domain (mCCP1, aa residue 137–371), but not the CCP1 precursor (preCCP1, aa residue 1–371). Furthermore, when co-expressed with CCP1, NIa-Pro is found to be co-localized with CCP1 in the lytic vacuole during the later stage, thereby effectively inhibiting CCP1 activity. This observation aligns with prior findings pertaining to the *U. maydis* effector Pit2 and maize cystatin CC9, both of which interact with PLCP protease domains [12,20]. Similarly, the Huanglongbing (HLB)-associated bacterium *Candidatus Liberibacter asiaticus* (CLas) effector SDE1 targets citrus PLCPs through interactions with conserved protease domains [23]. A most recent study uncovered the crucial role of PLCP TaRD21A in wheat against wheat yellow mosaic virus (WYMV, genus *Bymovirus* in the family *Potyviridae*) through the release of a small peptide, while the NIa protein encoded by WYMV suppresses the activity of TaRD21A to facilitate infection in susceptible wheat [86]. Thus, direct binding of effector proteins to plant PLCP protease domains appears to be a recurrent tactic employed by microbial pathogens to suppress the activity of these immune proteases.

PLCPs possess a conserved catalytic triad composed of three amino acid residues: Cys, His, and Asp. We observed that a mutant CCP1, with all the catalytic triad residues substituted, not only lost its protease activity but also had no ability to bind with NIa-Pro. This highlights the importance of the catalytic triad in the interaction between CCP1 and NIa-Pro. One theory posits that NIa-Pro impedes CCP1 protease activity by targeting this catalytic triad. Another theory proposes that CCP1 identifies NIa-Pro as a substrate, which would explain the requirement for an active catalytic triad for binding. Our data support that NIa-Pro could bind to CCP1, but cannot cleave CCP1, thus obstructing the CCP1 catalytic triad. This idea also resonates with the *U. maydis* Pit2 effector, regarded as a "false substrate", which suppresses the activity of various PLCPs [22]. In addition, it appears that the protein stability of NIa-Pro in samples co-expressing CCP1 and NIa-Pro is seemingly affected, implying the potential degradation of NIa-Pro as a substrate of CCP1 (S21 Fig). However, further investigation is required to confirm this possibility.

We further found that the 48 aa inhibitor motif is sufficient for function of NIa-Pro to interact with CCP1 and suppress its activity (Fig 6). The predicted structure of NIa-Pro also suggested that the inhibitory motif is surface-localized, potentially facilitating its association with CCP1. Evidence for the function of the 48 aa inhibitor motif is given by the mutational analyses that resulted in: i) loss of NIa-Pro–CCP1 interaction in LCI, ii) loss of protease inhibition of recombinant protein in the *in-vitro* assays and iii) loss of virulence, i.e. a SCMV mutant pSCMV-NIa-Pro$_{IMD}$-GFP which the 48 aa inhibitory motif was deleted. This line of evidence is further supported by the finding that the inhibitor efficiency of 48 aa inhibitor motif is higher than full length NIa-Pro. Amino acid sequence alignment of NIa-Pros belonging to the different potyviruses shows about 74.44% identity and there are some conserved residues in the inhibitory motif. Our further study indicated that among the five residues, only two residues K230 and D234 were crucial for the NIa-Pro-CCP1 interaction. The mutant NIa-Pro$_{K230A, D234A}$ resulted in the loss of inhibitor activity of NIa-Pro, suggesting that K230 and D234 were crucial for the function of NIa-Pro. Strikingly, we found that a mutant SCMV-NIa-Pro$_{K230A, D234A}$-GFP also showed the significantly reduced virulence compared to WT SCMV.

To our knowledge, this is the first report of viral-encoding protease utilize the inhibitory motif to suppress PLCPs, a hub in host immunity to promote virus infection. The most intriguing questions are about whether the inhibitor motif of NIa-Pro could be processed upon SCMV infection, and how the inhibitor motif is transported and targets the activated PLCPs to function.

Our study provide evidence that CCP1 plays positive roles in maize resistance to SCMV infection. In the early stages of virus infection, we found that both transcriptional and protein levels of CCP1 show an increasing trend, and the overall enzymatic activity of PLCPs increases. This correlates with our previous findings that SCMV infection leads to an increase in SA accumulation. The virus counters SA-mediated resistance by interacting with and inhibiting the enzymatic activity of key PLCPs involved in SA pathways. This is consistent with the observed inhibition of PLCP activity in the later stages of virus infection, allowing the virus to infect efficiently. This suggests that CCP1 does not trigger PCD. Since SCMV infection in maize plants results in mosaic symptoms but not leaf cell death, there might be a balance between the upregulation of CCP1 due to SCMV infection and its inhibition by SCMV NIa-Pro. Altogether, these discoveries unveil the significant involvement of plant PLCPs in fighting viral infections and uncover a previously undisclosed mechanism of virulence employed by potyviruses that infect plants.

## Materials and methods

### Plant growth and virus inoculation

Maize inbred line B73 and *N. benthamiana* plants were grown inside growth chambers maintained at 24/22˚C (day/night) and with a 16/8 h (light/dark) photoperiod. SCMV-BJ isolate is from a previously described source [52] and propagated in maize plants for further uses. SCMV inoculum was prepared by homogenizing SCMV-BJ-infected maize leaf tissues in a 0.01 M phosphate buffer (PB), pH 7.0, immediately prior to inoculation to 8-day-old maize seedlings as described [87]. Maize seedlings inoculated with PB were used as the mock-inoculated controls.

### Plasmid constructions

For CMV-based VIGS, a 210 bp fragment representing a partial sequence of *CCP1* (S2 Fig) was RT-PCR amplified and inserted into the pCMV201-2b$_{N81}$ vector [64] to generate pCMV201-CCP1 for the CMV-based VIGS.

Full-length sequence of *CCP1* (*Zm00001d016446*) was retrieved from the maize genome database (AGPv4, http://ensembl.gramene.org/Zea_mays/Info/Index) and then RT-PCR amplified using gene specific primers and a total RNA sample isolated from maize inbred line B73. For Y2H assays, we amplified SCMV open reading frames (ORFs) encoding the P1, HC-Pro, P3, P3N-PIPO, 6K1, CI, 6K2, VPg, NIa-Pro, NIb or CP, and cloned them individually into the pGBKT7 vector (Clontech, Mountain View, USA) to produce pBD-P1, pBD-HC-Pro, pBD-P3, pBD-P3N-PIPO, pBD-6K1, pBD-CI, pBD-6K2, pBD-VPg, pBD-NIa-Pro, pBD-NIb and pBD-CP, respectively. We then RT-PCR amplified the gene sequences encoding the precursor of CCP1 (preCCP1), the immature form of CCP1 (iCCP1, without the coding region for the signal-peptide) and the mature protease domain of CCP1 (mCCP1), and cloned them individually into the pGADT7 vector to produce pAD-preCCP1, pAD-iCCP1 and pAD-mCCP1, respectively.

For CoIP assay in *N. benthamiana* leaves, we first introduced three amino acid (aa) substitutions into the catalytic triad in mCCP1 to produce mCCP1$_{C161A, H303A, N330A}$. We then inserted the sequences encoding NIa-Pro, preCCP1, mCCP1, mCCP1$_{C161A, H303A, N330A}$ into

the pGD vector [88] to produce pGD-3Flag-NIa-Pro, pGD-preCCP1-3Myc, pGD-mCCP1-3Myc and pGD-mCCP1$_{C161A, H303A, N330A}$-3Myc, respectively. For LCI assay, the sequences encoding mCCP1, mCCP1$_{C161A, H303A, N330A}$ were amplified and individually inserted into the pCAMBIA1300-Nluc vector [89] to generate Nluc-mCCP1 and Nluc-mCCP1$_{C161A, H303A, N330A}$. And we then amplified the sequence encoding SCMV NIa-Pro, protease domain of NIa-Pro (pdNIa-Pro), inhibitor motif of NIa-Pro (imNIa-Pro) and several mutants (imNIa-Pro$_{W199A}$, imNIa-Pro$_{W203A}$, imNIa-Pro$_{N206A}$, imNIa-Pro$_{K230A}$, and imNIa-Pro$_{D234A}$) and cloned them into the pCAMBIA1300-Cluc to produce Cluc-NIa-Pro, Cluc-pdNIa-Pro, Cluc-imNIa-Pro, Cluc-imNIa-Pro$_{W199A}$, Cluc-imNIa-Pro$_{W203A}$, Cluc-imNIa-Pro$_{N206A}$, Cluc-imNIa-Pro$_{K230A}$, and Cluc-imNIa-Pro$_{D234A}$. For heterologous expression in *N. benthamiana* leaves, we cloned into the pGD vector to generate pGD-preCCP1-3Flag. For the subcellular localization assay in *N. benthamiana* leave cells, the full-length CDS of NIa-Pro was cloned into pCambia35S-GFP vector, resulting in plasmid pCambia35S-NIa-Pro-GFP, and the full-length CDS of CCP1 was cloned into pGD-mRFP, resulting in plasmid pGD-CCP1-mRFP.

For heterologous expression in *E. coli*, we first amplified the sequences encoding mCCP1 or mCCP1$_{C161A, H303A, N330A}$, and cloned them individually into the pCold-TF vector (TaKaRa, Dalian, China) to generate pCold-TF-mCCP1 and pCold-TF-mCCP1$_{C161A, H303A, N330A}$. We then amplified the sequence encoding SCMV NIa-Pro and cloned it into the pGEX vector (Invitrogen, Carlsbad, CA, US) to produce pGEX-NIa-Pro$_{SCMV}$.

The infectious clones of two mutant viruses pSCMV-NIa-Pro$_{K230A, D234A}$-GFP and pSCMV-NIa-Pro$_{IMD}$-GFP were constructed by utilizing the infectious clone pSCMV-GFP for mutagenesis. The full-length CDS of 3Flag-NIa-Pro was inserted into FoMV-mediated overexpression vector (pV101) [69], resulting in plasmids pV101-3Flag-NIa-Pro.

All the constructs were sequenced before use. Sequences of primers used in this study are listed in S2 Table.

## CMV-based VIGS in maize

The CMV-based VIGS assay was performed as reported previously [64]. Agrobacterium cultures harboring pCMV101, pCMV301, and pCMV201-CCP1 or pCMV201-GUS were mixed at a 1:1:1 ratio and then infiltrated into *N. benthamiana* leaves. At 4 dpai, crude extracts from the infiltrated leaves were used to inoculate maize B73 seeds using a vascular puncture inoculation method [64].

## FoMV-mediated gene over-expression in maize plants

Inoculation of *N. benthamiana* and maize seedlings with the foxtail mosaic virus (FoMV) expression constructs was performed as described previously [69]. First, agrobacterium cultures harboring FoMV-3Flag-GFP or FoMV-3Flag-NIa-Pro were infiltrated into *N. benthamiana* leaves. At 4 dpi, infiltrated leaves were ground in 10% (w/v) 0.01 M phosphate buffer, supplemented with 1% (w/v) Celite 545 AW (Sigma-Aldrich). Then crude extracts were used for rub inoculation of maize seedlings. At 10 min post inoculation, leaves were sprayed with tap water, and plants were bagged. Plants were maintained under high humidity and no light for 1 day and finally returned to standard growth conditions for subsequent experiments.

## Total RNA extraction and RT-qPCR analysis

Total RNA was isolated from various tissue samples using TRIzol reagent (Tiangen, Beijing, China) followed by a treatment with an RNase-free DNase I (TaKaRa, Dalian, China). First-strand cDNA (20 μl) was synthesized using 2.0 μg total RNA and an oligo (dT) primer. Ten-fold diluted cDNA samples, gene-specific primers and a FastSYBR kit (CWBIO, Beijing,

China) were used for qPCR on an ABI 7500 Real Time PCR system (Applied Biosystems, Foster City, CA, USA). The expression level of *ZmUbi* was used as an internal control and the relative expression of each gene was calculated using the $2^{-\Delta\Delta Cm}$ method [90]. Statistical differences between the treatments were determined using the Student's *t*-test.

## Immunoblotting assay and statistical analyze

Total protein preparation and separation through electrophoresis were the same as described [87]. Polyclonal antibody of SCMV CP or CCP1 was used at a dilution of 1:5,000. Monoclonal antibodies of actin and GST (CW0258 and CW0084, CWBIO) were used at a dilution of 1:5,000. Monoclonal antibodies of anti-His (CW0285, CWBIO), anti-Myc and anti-Flag (A5598 and A8592, Sigma-Aldrich, Taufkirchen, Germany) were used at a dilution of 1:10,000. Streptavidin-HRP conjugates (Thermo Scientific) were used at a dilution of 1:5,000. For further quantitative analysis, the accumulation levels of CCP1, SCMV CP, and active PLCPs were normalized to actin control.

## Heterologous expression in *E. coli* and Protein Purification

Plasmids pCold-TF-mCCP1, pCold-TF-mCCP1$_{C161A,H303A,N330A}$, and pGEX-NIa-Pro were individually transformed into *E. coli* strain BL21 (DE3) cells. Expression of recombinant protein were induced by addition of 0.5 mM isopropyl-β-D-thiogalactoside (IPTG) at 16˚C overnight. Recombinant proteins were purified following the manufacturer's instructions (TaKaRa, Dalian, China).

## Protein interaction assays

Y2H assay was performed using the Matchmaker Yeast Two-Hybrid System as instructed (Clontech Laboratories, Mountain View, CA, USA). Briefly, specific combinations of the AD and BD vectors were co-transformed into yeast cells and then allowed to grow on the SD/-Leu/-Trp DO (DDO) medium or the SD/–Leu/–Trp/–Ade/–His DO (QDO) medium at 30˚C. The pGADT7-T/pGBKT7-53 (AD-T/BD-53) plasmid was used as a positive control, and pGADT7-mCCP1/pGBKT7 (AD-mCCP1/BD) was used as a negative control.

For *in vivo* co-immunoprecipitation (Co-IP) assay, plasmid pGD-3Flag-NIa-Pro, pGD-mCCP1-3Myc or pGD-mCCP1$_{C161A,H303A,N330A}$-3Myc were individually transformed into Agrobacterium cells and then heterologously expressed in the leaves of 4-week-old *N. benthamiana* plants through infiltration. The infiltrated leaves were harvested at 72 hours post agro-infiltration (hpai) and grounded (1 g tissue per sample) in an extraction buffer (50 mM Tris-HCl, pH7.5, 150 mM NaCl, 1 mM EDTA, 0.1% Triton X-100 and 10% glycerol) supplemented with 1 mM phenylmethyl sulfonyl fluoride and a protease inhibitor cocktail as instructed (Sigma-Aldrich, Taufkirchen, Germany). The crude leaf extracts were centrifuged, filtered through 0.45-μm filters, and checked for proteins concentrations through the Bradford assay [91]. Equal amount of protein in a sample (about 1.2 ml) was mixed with 40 μl of anti-Flag M2 Affinity gel (A2220, Sigma-Aldrich) and then incubated for 3 h at 4˚C on a rotary incubator to immunoprecipitate the target protein. The precipitates were washed four times with an ice-cold IP buffer followed by four times with an ice-cold PBS buffer. The resulting samples were individually solubilized in a loading buffer, boiled for 5 min, and then analyzed by immunoblotting assay using specific antibodies.

For LCI assay, equal amounts of Agrobacterium cultures containing different cLUC and nLUC construct pairs were co-transformed into *N. benthamiana*. The plants were harvested at 48 hpai and sprayed with a 1 mM luciferin solution (Promega, Beijing, China), and then

examined and imaged using a low-light cooled charge-coupled-device camera (Lumazone 1300B, Teledyne Princeton Instruments).

For *in vitro* pull-down assay, GST-NIa-Pro, TF-mCCP1 or TF-mCCP1$_{C161A, H303A, N330A}$ was expressed in *E. coli* strain BL21 (DE3) cells followed protein purification. The purified TF-mCCP1, TF-mCCP1$_{C161A, H303A, N330A}$, GST and GST-NIa-Pro were mixed in specific combinations in glutathione resins (50 μl per reaction, Thermo Scientific, USA) and then incubated for 3 h at 4˚C. The resins were rinsed with an IP buffer containing 20 mM Tris-HCl, 200 mM KCl, 0.1 mM EDTA, 0.05% Triton X-100, pH6.0, boiled for 10 min in a 2 × SDS sample buffer, and then subjected to gel electrophoresis followed by immunoblotting using an anti-GST or an anti-His antibody.

## Protease activity assay

Plasmids including pGD-3Flag-mCCP1, pGD-3Flag-mCCP1$_{C161A,H303A,N330A}$ or pGD-3Flag-mSbCP1 were expressed heterologously in 4-week-old *N. benthamiana*. At 48–72 hpai, the harvested leaf samples (1 g per sample) were ground individually in liquid nitrogen and then homogenized in a 5 ml of ice-cold 50 mM Tris buffer, pH7.2, containing 0.2% polyvinylpyrrolidone (PVP) and 5 mM mercaptoethanol. Then, the resulting samples were applied to immunoprecipitate Flag-tagged proteins with addition of anti-Flag M2 Affinity gel as previously described in Co-IP assay. After immunoprecipitation, the enriched proteins were washed four times with the IP buffer and then twice in the PBS buffer.

Protease activity was measured using a fluorimetric substrate (Z-Phe-Arg-7-amido-4-methylcoumarin, Sigma-Aldrich), which leads to release of fluorescence at 460 nm when cleaved by protease activity [92]. Specific concentrations of purified TF-mCCP1 or the anti-Flag M2 Affinity Gel-enriched 3Flag-mCCP1 was mixed with the purified GST protein (negative control), GST-NIa-Pro or E-64 (positive control), and then incubated at 4˚C for 2 h. After incubation, each protein mixture (90 μl) was combined with 10 μl of 10 μM substrate followed by the measurement of absorbance at 460 nm using a microplate reader (SpectraMax i3x, Molecular Devices, USA).

## Prediction of protein structure

The structures of proteins were predicted based on the complete amino acid sequence, utilizing the online structure prediction software SWISS-MODEL (https://swissmodel.expasy.org/interactive) in combination with the PDB protein structure database (https://www.rcsb.org/).

## Activity-based protein profiling

For the total PLCPs activity in maize, total extracts from the 1st-systemic leaves of SCMV or mock-inoculated plants were incubated with a final concentration of 2 μM DCG-04 (Medkoo, USA) for 4 h at room temperature, followed by precipitation with 100% ice-cold acetone. Samples were centrifuged at 12,000×g, washed with 70% acetone, then centrifuged again. Precipitated products were re-suspended in 50mM Tris buffer (pH 6.4) and either used directly for immunoblotting using Streptavidin-HRP conjugates (Thermo Scientific).

For the semi-in vitro ABPP assay, total leaf extracts from CCP1-3Flag or CCP1$_{C161A,H303A, N330A}$-3Flag was pretreated with either 5 μM E-64 or 1.0 μM GST-NIa-Pro or GST proteins. After pretreatment, the samples were incubated with 2 μM DCG-04 for 4 h at room temperature, followed by 3 min centrifugation at 3,000 *g* and then two rinses in PBS buffer. The resulting protein samples were individually mixed with a 2 × SDS sample loading buffer, boiled for 10 min, and then analyzed through immunoblotting assays using a Streptavidin-HRP conjugate as instructed.

The experiments were repeated two times with similar results.

## Phylogenetic analysis

Multiple amino acid sequences were aligned using the Clustal X software [93]. MEGA v6.06 was used to construct the Neighbor-Joining phylogenetic tree using the Poisson model and a bootstrap value of 100 [94].

## Supporting information

**S1 Data. Excel spreadsheet containing, in separate sheets, the underlying numerical data and statistical analysis for Figs 1A, 1D, 1E, 2B, 2E, 3B, 3C, 3D, 5A, 5D, 7B, 7C, 8A, 8C, 8E, 8F, S1A, S1B, S3, S4, S5A, S5B, S5C and S5D.**
(XLSX)

**S1 Fig. An RNA sequencing analysis shows that the expression of *CCP1* was significantly up-regulated following SCMV infection.** A) Relative expression of maize *PLCPs* at 3 days post inoculation (dpi). B) Relative expression of maize *PLCPs* at 9 dpi.
(TIF)

**S2 Fig. Alignment using multiple maize *PLCP* genes nucleotide sequences.** The redlined nucleotide sequence is the partial *CCP1* sequence inserted into the CMV-based VIGS vectors for specific silencing of *CCP1*.
(TIF)

**S3 Fig. The contents of JA, ABA and IAA in the CMV-CCP1-inoculated or the CMV-GUS-inoculated maize plants.**
(TIF)

**S4 Fig. RT-qPCR analysis of the relative accumulation levels of CMV genomic RNA in the SCMV-infected CMV-CCP1 maize plants at 10 dpi.** The *coat protein* (*cp*) gene of CMV RNA3 was used for RT-qPCR analysis. The statistical results evaluated by Student's t test analysis indicate that the differences between the two are not significant (ns, $P > 0.05$).
(TIF)

**S5 Fig. Over-expression of CCP1 in maize protoplasts suppresses SCMV RNA replication.** Protoplasts were separately co-transfected with a mixture of pGD-CCP1-mRFP and SCMV RNA, a mixture of pGD-mRFP and SCMV RNA, or a mixture of pGD-CCP1$_{C161A,H303A,N330A}$-mRFP and SCMV RNA. The expression level of *CCP1* (A), the relative accumulation of SCMV RNA (B) and the expression level of SA marker genes *ZmPR1* (C) and *ZmPR5* (D) were determined at 18 h post transfection. Three independent experiments were conducted with at least three biological replicates per treatment. Error bars are the means ± SE. **, $P < 0.01$; ***, $P < 0.001$; ****, $P < 0.0001$; determined by the unpaired t-test.
(TIF)

**S6 Fig. Y2H assay to determine the interaction between NIa-Pro and preCCP1, iCCP1, or mCCP1.** Only mature protease domain of CCP1 (mCCP1) interacts with NIa-Pro, while other forms including precursor of CCP1 (preCCP1) and signal-peptide deletion CCP1 (iCCP1) do not interact with NIa-Pro in yeast cells. Yeast cells grown on the QDO selective medium are the cells with a positive protein–protein interaction. The AD-T+BD-53 serves as the positive control.
(TIF)

**S7 Fig. Co-immunoprecipitation (Co-IP) assay was conducted to verify the interaction between preCCP1 and NIa-Pro in *N. benthamiana* leaves.** 3Flag-NIa-Pro was co-expressed with preCCP1-3Myc or GUS-3Myc in *N. benthamiana* leaves through agro-infiltration. Leaf tissues were harvested at 48 hours post agroinfiltration (hpai) for Co-IP analysis. Samples from plants co-expressing 3Flag-NIa-Pro and GUS-3Myc were used as a negative control. Co-IP assay was performed using an anti-Flag affinity agarose gel. Protein samples (Input) and the immunoprecipitated protein samples (Flag-IP) were analyzed through immunoblotting assays using an anti-Flag or an anti-c-Myc antibody. *, the protein bands corresponding to the expected 3Flag-NIa-Pro, GUS-3Myc or preCCP1-3Myc.
(TIF)

**S8 Fig. Luciferase (Luc) complementation imaging (LCI) analysis of the interaction between NIa-Pro and preCCP1.** The indicated plasmid pairs were transiently co-expressed in *N. benthamiana*. Then the luminescent signal was collected at 48 hpai.
(TIF)

**S9 Fig. Y2H assay shows that NIa-Pro does not interact with inactivated mCCP1$_{C161A, H303A,N330A}$.** In mCCP1$_{C161A,H303A,N330A}$, the catalytic triad residues were replaced by Alanine. Yeast cells grown on the QDO selective medium are the cells with a positive protein–protein interaction. The AD-T+BD-53 serves as the positive control.
(TIF)

**S10 Fig. LCI assay showes that NIa-Pro does not interact with inactivated mCCP1$_{C161A, H303A,N330A}$.** In mCCP1$_{C161A,H303A,N330A}$, the catalytic triad residues were replaced by Alanine. The indicated plasmid pairs were transiently co-expressed in *N. benthamiana*. Then the luminescent signal was collected at 48 hpai.
(TIF)

**S11 Fig. NIa (VPg-NIa-Pro) can also interact with mCCP1 in yeast cells.** The plasmids pGADT7-mCCP1 and pGBKT7-NIa, or pGADT7-mCCP1$_{C161A, H303A, N330A}$ and pGBKT7-NIa were co-transformed into yeast. The co-transformants were grown on the selective medium at 30°C for 3–4 d. Yeast cells grown on the QDO selective medium are the cells with a positive protein–protein interaction. The AD-T+BD-53 serves as the positive control.
(TIF)

**S12 Fig. Three-dimensional model of mCCP1 (A) and mutant mCCP1$_{C161A,H303A,N330A}$ (B), in cartoon style.** The N and C termini of the protein are indicated by black arrows, and the catalytic triad key residues are labeled as yellow (C161), red (H303), and green (N330).
(TIF)

**S13 Fig. The localization pattern of SCMV NIa-Pro.** Confocal images of *N. benthamiana* leaf cells expressing NIa-Pro-GFP or GFP at 48 hpai. DAPI staining indicates the nucleus of tobacco cells, revealing that the green fluorescence of NIa-Pro-GFP is localized not only in the cytoplasm but also in the nucleus. Images (left to right) showed GFP fluorescence, DAPI fluorescence, bright field and overlay of the three images. Scale bars, 10 μm.
(TIF)

**S14 Fig. The presence of intact fusion protein NIa-Pro-GFP in *N. benthamiana* leaf cells was assessed by immunoblotting analysis.** At 48 hpai, infiltrated leaves were sampled for immunoblotting assays using an anti-GFP antibody.
(TIF)

**S15 Fig. CCP1 colocalizes with the lytic vacuolar marker AALP (A) and the endoplasmic reticulum marker HDEL (B) in the epidermal cells of *N. benthamiana* leaf.** Images (left to right) showed CFP or GFP fluorescence, RFP fluorescence, bright field and overlay of the three images. Scale bars, 50 μm.
(TIF)

**S16 Fig. SCMV NIa-Pro co-localizes with full-length of CCP1.** Confocal images of *N. benthamiana* leave cells co-expressing CCP1-mRFP and NIa-Pro-GFP at 48 (A) and 72 hpai (B). C) The overlay of CCP1-mRFP and NIa-Pro-GFP in the cytoplasm of maize cell at 16 h after bombardment. Images (left to right) showed RFP fluorescence, GFP fluorescence, bright field and overlay of the three images. Scale bars, 50 μm.
(TIF)

**S17 Fig. NIa-Pro did not affect the stability of CCP1 in planta.** pGD-preCCP1-3Myc or pGD-mCCP1-3Myc was co-infiltrated with pGD-GFP plus pGD-3Flag-NIa-Pro or pGD-3Flag-GUS (control) in *N. benthamiana* leaves. At 3 dpi, infiltrated leaves were sampled and analyzed for the accumulation levels of preCCP1-3Myc (A) and mCCP1-3Myc (B) via immunoblotting assays. The strength of detection signal was analyzed using the ImageJ software. The red asterisk indicates the protein band corresponding to preCCP1- or mCCP1-3Myc. Three independent experiments were conducted with at least three biological replicates per treatment.
(TIF)

**S18 Fig. Multiple sequence alignment of potyviruses-encoded NIa-Pro.** The following viruses were included in the analysis: soybean mosaic virus (SMV, AAB22819.2), watermelon mosaic virus (WMV, AAA48497.2), dasheen mosaic virus (DaMV, NP_613274.1), papaya leaf distortion mosaic virus (PLDMV, NP_870995.1), papaya ringspot virus (PRSV, AAG47346.1), yam mosaic virus (YMV, QGA88722.1), lettuce mosaic virus (LMV, QEG79196.1), plum pox virus (PPV, CVK35891.1), potato virus Y (PVY, AKG94974.1), turnip mosaic virus (TuMV, QBQ58061.1), japanese yam mosaic virus (JYMV, NP_051161.1), potato virus A (PVA, YP_006395324.1), tobacco etch virus (TEV, ABJ16044.1), tobacco vein mottling virus (TVMV, CAA27720.1), onion yellow dwarf virus (OYDV, NP_871002.1), sugarcane mosaic virus (SCMV, AMM72620.1), pennisetum mosaic virus (PenMV, YP_006395348.1), sorghum mosaic virus (SrMV, CAX36858.1), maize dwarf mosaic virus (MDMV, NP_569138.1) and johnsongrass mosaic virus (JGMV, ALS88434.1). The regions highlighted by the pink bounding box are hypothesized to represent the functional peptide of the NIa-Pro C-terminus. The amino acid residues marked by pentagram symbols represent the selected functional amino acids for mutagenesis.
(TIF)

**S19 Fig. Y2H assay shows that residues K230 and D234 are both vital for the NIa-Pro–CCP1 interaction.** Yeast cells grown on the QDO selective medium are the cells with a positive protein–protein interaction. The AD-T+BD-53 serves as the positive control.
(TIF)

**S20 Fig. The expression of several SCMV-encoded proteins in SCMV-GFP- or its double mutant variants SCMV-NIa-Pro$_{K230A, D234A}$-GFP-infected maize leaves.** At 7 dpi, leaf samples were collected for immunoblot analysis using specific antibodies to detect the expression of SCMV HC-Pro (A), CI (B), NIa-Pro (C), and CP (D).
(TIF)

**S21 Fig. Full-length of CCP1 (preCCP1) affects the stability of NIa-Pro in planta.** pGD-preCCP1-3Myc or pGD-mCCP1-3Myc was co-infiltrated with pGD-3Flag-NIa-Pro in *N. benthamiana* leaves. pGD-GUS-3Myc co-expressed with pGD-3Flag-NIa-Pro was used as control. At 3 dpi, infiltrated leaves were sampled and analyzed for the accumulation levels of 3 Flag-NIa-Pro via immunoblotting assays. The strength of detection signal was analyzed using the ImageJ software. Three independent experiments were conducted with at least three biological replicates per treatment.
(TIF)

**S1 Table. The expression of *PLCP* genes was modulated by diverse plant viruses.**
(DOCX)

**S2 Table. Primers used in this study.**
(DOCX)

**S1 Information. The sequencing analysis of the SCMV mutant prior to the inoculation of maize.**
(PPTX)

## Author Contributions

**Conceptualization:** Wen Yuan, Tao Zhou.

**Data curation:** Wen Yuan, Xi Chen.

**Formal analysis:** Wen Yuan, Xi Chen.

**Funding acquisition:** Tao Zhou.

**Investigation:** Wen Yuan, Kaitong Du, Yanyong Cao, Zaifeng Fan.

**Methodology:** Wen Yuan, Tong Jiang, Mengfei Li, Yanyong Cao.

**Project administration:** Wen Yuan, Xiangdong Li, Tao Zhou.

**Resources:** Wen Yuan, Xiangdong Li, Zaifeng Fan, Tao Zhou.

**Software:** Wen Yuan.

**Supervision:** Zaifeng Fan, Tao Zhou.

**Validation:** Tao Zhou.

**Visualization:** Tao Zhou.

**Writing – original draft:** Wen Yuan.

**Writing – review & editing:** Gunther Doehlemann, Tao Zhou.

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
