## [Decision Letter · Decision Letter 0]

16 Oct 2023

Dear Dr. ZHOU,

Thank you very much for submitting your manuscript "NIa-Pro of sugarcane mosaic virus targets Corn Cysteine Protease 1 (CCP1) to undermine salicylic acid-mediated defense in maize" for consideration at PLOS Pathogens. As with all papers reviewed by the journal, your manuscript was reviewed by members of the editorial board and by several independent reviewers. In light of the reviews (below this email), we would like to invite the resubmission of a significantly-revised version that takes into account the reviewers' comments.

The reviewers found your work interesting, and addressing unexplored research territory. However, all three reviewers raised significant major points which need to be addressed in your point-by-point response.

Please pay particular attention to the criticism about the cell biology and the colocalisation of the the proteins raised by reviewer 3.

Your revision will also require western blot experiments showing that the fusions are intact and the inclusion of free GFP as a negative control.

Please provide uncropped original gels as supplementary where the cropped area is indicated as suggested by reviewer 1.

The points raised by reviewer 2 should be considered for a revision and integrated in your discussion if you cannot carry out additional experiments.

We cannot make any decision about publication until we have seen the revised manuscript and your response to the reviewers' comments. Your revised manuscript is also likely to be sent to reviewers for further evaluation.

Sincerely,

Sebastian Schornack, Ph.D.

Academic Editor

PLOS Pathogens

Sonja Best

Section Editor

PLOS Pathogens

Kasturi Haldar

Editor-in-Chief

PLOS Pathogens

orcid.org/0000-0001-5065-158X

Michael Malim

Editor-in-Chief

PLOS Pathogens

orcid.org/0000-0002-7699-2064

The reviewers found your work interesting, and addressing unexplored research territory. However, all three reviewers raised significant major points which need to be addressed in your point-by-point response.

Please pay particular attention to the criticism about the cell biology and the colocalisation of the the proteins raised by reviewer 3.

Your revision will also require western blot experiments showing that the fusions are intact and the inclusion of free GFP as a negative control.

Please provide uncropped original gels as supplementary where the cropped area is indicated as suggested by reviewer 1.

The points raised by reviewer 2 should be considered for a revision and integrated in your discussion if you cannot carry out additional experiments.

Reviewer's Responses to Questions

**Part I - Summary**

Reviewer #1: This manuscript addresses an interesting and relatively unexplored question: what is the role plant papain-like cysteine proteases in antiviral defense responses, and how do viruses counteract these defense responses. The authors provide convincing evidence for a role of CCP1 in maize antiviral responses to SCMV, by (a) showing activation of CCP1 expression in response to SCMV infection and (b) demonstrating that knock-down of this gene leads to increased virus infection and reduced SA-dependent responses. Next, they show that the viral NIa-Pro interacts with CCP1 and compromises its activity, suggesting a viral counter-defense response. The manuscript is generally well-written and follows a logical flow. The conclusions are generally well-supported by the experimental results. Below a list of concerns that need to be addressed, in particular proper labeling of the gels with molecular mass markers and ensuring that names of constructs/proteins are consistent throughout the text and figures and that conclusions are carefully worded.

Reviewer #2: Host proteases play important roles in both plant defense and viral infection. In this manuscript, Yuan et al. investigated on how potyviral NIa-pro targets host protease CCP1 to dampen SA-mediated defense in maize. CCP1 in maize SA signaling has been revealed years ago, and how viral proteins target CCP1 to suppress host defense and promote viral infection remains elusive. Whilst this work is intriguing, I have issues with the robustness and interpretation of the presented data as outlined in the following:

Reviewer #3: This is an interesting, and well-written manuscript on the discovery of Nla-Pro of sugarcane mosaic virus as an inhibitor of papain-like protease CCP1 of maize. Overall the science is sound and the conclusions are supported by experiments.

**Part II – Major Issues: Key Experiments Required for Acceptance**

Reviewer #1: 1. One of the main concerns is that none of the gels show the migration of molecular mass markers. Original uncropped gels are not included. Thus, it is very difficult to independently assess that the labeled bands do indeed correspond to specific proteins (also because the calculated molecular mass of the different proteins is not provided). This is particularly a concern in some figures in which unlabeled additional bands are noted. For example, what is the unlabeled predominant lower band in the His Abs blot (pull-down) in Fig 4E? All gels from ALL figures, need to be labeled with the migration of molecular mass markers so that the reader can assess the migration of all bands in relation to the calculated molecular mass of the expected proteins. I also noted a probable labeling error in Fig S5, where the first lane should be positive for GUS 3Myc and NIa-Pro 3Flag but not for preCCP1 3 Myc (otherwise the relative migration of the bands do not make sense).

2. For all quantification analyses (levels of various proteins, activity of CCP1), a gel with the levels of the control actin protein is presented. Were the actin levels taken into account when calculating relative concentrations of CCP1, viral CP as well as the activity of CCP1? This is not clearly explained in the methods. Please provide more details.

3. The impact of the double mutation (K230A, D234A) is not completely clear (lines 420-427). Although the mutations prevent interaction of NIa-Pro with CCP1 (Fig. 6D) and the inhibition of CCP1 activity (Fig 6E), they do not impact the level of infection of the infectious clone in N benthamiana (Fig 7A-B), but do impact infection in maize after re-inoculation of the N benthamiana infected extracts (Fig 7D-E). Please explain better these discrepancies and rephrase the conclusions more carefully. In fact, I am wondering if the virus sequence was confirmed after passage in N benthamiana and before reinoculation of maize, to confirm that there were no additional mutations that may have been introduced upon passage in the first host.

Reviewer #2: 1. Authors state that CCP1 plays a role in modulating SA signaling via knocking down this gene. Moreover, NIa-Pro suppresses CCP1’s protease activity to dampen plant defense. Whether CCP1’s protease activity is important for SA signaling and plant defense against SCMV infection has bot been investigated here, although similar mechanism has been reported in maize previously. These two parts seems unrelated currently. Authors should use this mutant mCCP1C161A, H303A, N330A to further validate this question. Also, can CCP1 directly target SCMV proteins for cleavage to suppress SCMV infection?

2. NIa-Pro interacts with CCP1, but how this interaction affects CCP1’s function, especially its protease activity, is still largely unknown. Why NIa-Pro? Subcellular localization or protein accumulation change? Actually, we can see CCP1 protein accumulation is indeed decreased under SCMV infection at early infection stages shown in Fig.1B.

3. Truncated NIa-Pro and its mutants can inhibit SCMV infection, which may not be caused by its deficit in suppressing CCP1 activity. As NIa-Pro is the main protease for potyviral polyprotein processing and also has other important functions, authors should test whether these NIa-Pro mutants used in Fig.7 still harbour normal protease activity et al. Also, SA level, PLCP enzyme activity and CCP1 expression should be investigated in plants infected with SCMV mutants compared with WT SCMV.

4. The effect of NIa-pro and these mutants on plant defense needs to be investigated. Also, if NIa-Pro indeed target CCP1 to dampen SA-mediated host defense, genetic evidences are lacking.

Reviewer #3: The cell biology should be supported by western blot experiments showing that the fusions are intact. The Nla-Pro-GFP fusion shows a similar localization as free GFP would have, which is missing as a negative control. Likewise, it is hard to imagine that CCP1 would be in the cytoplasm if it has a signal peptide for secretion. Please provide stronger proof that these proteins co-localise and discuss possible mechanisms.

**Part III – Minor Issues: Editorial and Data Presentation Modifications**

Reviewer #1: 1. Lines 137-138, based on Fig. 1E, the surge of PLCP activity was only significantly at 7 dpi and not at 12 dpi. Also, line 140, since the impact of SCMV infection is dependent on the timing of infection, rather than stating that SCMV infection “amplifies” maize PLCPs, it is more accurate to state that it “alters” the regulation of maize PLCPs.

2. Line 84-85: This sentence needs references.

3. Lines 196-199. It may be useful to explain what forms of CCP1 are normally found in plants. I am assuming that the mature protease is normally released from the precursor by autocatalytic cleavage. Has this been demonstrated (either for this protease or for related proteases)? If so, please provide references. What are the biological functions of the precursor and mature forms of CCP1?

4. Lines 218-219. It is also possible that the triple mutation of the catalytic triad changed the overall conformation of the protein and altered the presentation of another interacting domain.

5. Lines 270-272. This part of the text is not very clear, what is meant by “expressed fewer heterologous proteins”? What are those heterologous proteins?

6. Lines 284: the C-terminal domain is introduced as an inhibitor motif. Is it related to other known inhibitor motifs, or is this statement only based on experiments that are described later in the manuscript? If there was no previous known inhibitor activity of this C-terminal domain, it may be better to state that the C-terminal domain has an unknown function but could potentially act in the CCP1 inhibition since it is surface-localized. Is the C-terminal inhibitor domain also present in other 3C-like proteases from plant or animal picorna-like viruses or is it specific to potyviruses?

7. Lines 295 and Fig S12 (also lines 420-423). These sections of the text are not clear. In Fig S12, five amino acids are highlighted as being completely conserved in all NIa-Pros. The reader assumes that the five amino acids mutated correspond to the strictly conserved amino acids, but this is not the case. Residues N206, K230 and D234, are not strictly conserved in all potyvirus NIa-Pro, but were chosen for the mutagenesis analysis and the last two did show an impact on the ability of NIa-Pro to interact with CCP1. All mutated residues should be labeled in Fig S12 and a better explanation should be provided as to why these amino acids were chosen for the mutagenesis analysis.

8. Lines 301-303. I could not figure out which lane corresponds to the in vitro synthesized 48 aa inhibitor peptide in Fig. 6E. Do the authors mean that the inhibitor domain was fused with the GST (construct GST-imNIa-Pro?)

9. Lines 316-318, change “areas expressing SCMV” for “areas infected with SCMV” (two instances)

10. Line 408, it is stated that NIa-Pro binds but cannot cleave CCP1. The sentence is not clear. Is it meant that CCP1 does not cleave NIa-Pro to release it from the catalytic site of CCP1, or is it meant that NIa-Pro does not cleave CCP1.

11. Lines 433-445. Please refer to Fig 8 and explain in the text that this is a proposed model.

12. Fig. 1 legend, 3rd line. Replace “phosphorate” by “phosphate”

13. Fig. 3 legend. Please explain what the 4 stars represent in terms of P values.

14. Fig. 4 legend. Please explain what the AD-T+BD-53 constructs are. Positive control?

Reviewer #2: 1. In fig.1A and B, relative RNA and protein accumulation levels of CCP1 are not consistent between two treatments, and even contrary. Also, the PLCP activities is not consistent with CCP1 accumulation level. For example, under SCMV infection at 7 dpi, CCP1 is significantly downregulated (1B), however, PLCP activities are upregulated (1D), compared with mock. The quantified results in Fig.1E are also not consistent with 1D (10 and 12 dpi). CCP1 antibody and gene primer may be not specific, how about CCP2? Authors should provide immunoblot results against SCMV CP in Fig.1B.

2. In fig.2, Authors use CMV-mediated VIGS system to investigate the effect of silencing CCP1 on the second SCMV infection. Since silencing of CCP1 suppressed plant defense, thus, we can speculate that CMV accumulation has been also increased and then SCMV infection in CCP1-silenced plants. Authors should exclude this possibility.

3. Authors should also test the effect of CCP1 over-expression on SA signaling and SCMV infection in maize protoplasts or leaves.

4. In Fig.4B, authors use nuclear Y2H system to investigate CCP1 interaction with each of SCMV proteins. Since most of SCMV proteins are membrane and cytoplasm protein, nuclear Y2H is not appropriate.

4. Lines 202-203: 3×myc, 3×flag. If use partial gus fragment, authors should indicate.

5. Line 421: about-identity?

6. Fig.6A, P3N-PIPO, not pipo.

7.In Fig.6, authors should also use other methods including co-ip to confirm the protein-protein interaction results. Only LIC assay data is not solid.

Reviewer #3: Please explain Nla-Pro and Vpg in the introduction.

Please explain to which PLCP class CCP1 belong to.

All gene names (e.g. CCP1 in L118) should be printed in italics.

Please state where DCG-04 was obtained from.

I don’t understand that the PLCP activity goes up when there is less CCP1 at 7dpi in Fig1, and why you do not see the suppression of PLCP activity during SCMV infection.

L154 should read: ‘CMV-GUS control plants’.

Please check if the catalytic mutant of CCP1 accumulates as its precursor on a western blot. That would explain why it cannot interact with Nla-Pro.

L226 I am not convinced that the figures demonstrate that CCP1 is in the cytoplasm. Better images and plasmolysis would be required, supported by WB to show that the fusion is intact.

Why does FoMV-3Flag-Nia-Pro expressing fewer heterologous proteins? Are you not comparing two different proteins having the same tag?

Please explain what type of virus CMV and FoMV are. If they are also potex viruses then this might interfere in SCMV infection.

How was the structure predicted for Fig6B?

Please add a reference for the C-terminal inhibitor motif (L284).

Please add explain how likely it is that previously studied PLCPs are not co-silenced in CMV-CCP1 plants.

Are the first two bars in Fig2D and 2F based on the same dataset? This should be disclosed in the legend. Or better would be to remove this from Fig2F to avoid confusion.

Fig3BCD The stats should be added to compare CMV-GUS with CMV-CCP1.

Were constructs for Fig4 produced with or without signal peptide?

Please add stats to support Fig5B and 5C.

PLOS authors have the option to publish the peer review history of their article (what does this mean?). If published, this will include your full peer review and any attached files.

Reviewer #1: **Yes: **Hélène Sanfaçon

Reviewer #2: No

Reviewer #3: No
---

## [Decision Letter · Decision Letter 1]

17 Feb 2024

Dear Dr. ZHOU,

Thank you very much for submitting your manuscript "NIa-Pro of sugarcane mosaic virus targets Corn Cysteine Protease 1 (CCP1) to undermine salicylic acid-mediated defense in maize" for consideration at PLOS Pathogens. As with all papers reviewed by the journal, your manuscript was reviewed by members of the editorial board and by several independent reviewers. The reviewers appreciated the attention to an important topic. Based on the reviews, we are likely to accept this manuscript for publication, providing that you modify the manuscript according to the review recommendations and address constructively all comments, in particular the concern of reviewer 3.

Sincerely,

Sebastian Schornack, Ph.D.

Academic Editor

PLOS Pathogens

Sonja Best

Section Editor

PLOS Pathogens

Michael Malim

Editor-in-Chief

PLOS Pathogens

orcid.org/0000-0002-7699-2064

Reviewer Comments (if any, and for reference):

Reviewer's Responses to Questions

**Part I - Summary**

Reviewer #1: The new version of the manuscript is significantly improved and I must commend the authors for their thorough and detailed response to the comments. All my original concerns have been addressed to satisfaction. In reading the revised version, I found a few minor mistakes that will need to be corrected before publication.

Reviewer #2: The authors have addressed most of my concerns. I am satisfied with the responses, although there are still some minor concerns required to be revised as follow.

Reviewer #3: This manuscript has been greatly improved when compared to the previous version and most of my issues have been addressed. I am particularly glad that the CCP1 subcellular localisation is addressed experimentally.

**Part II – Major Issues: Key Experiments Required for Acceptance**

Reviewer #1: (No Response)

Reviewer #2: (No Response)

Reviewer #3: The authors claim that CCP1 was not expressed with a signal peptide for the experiments in Fig4, but Fig4CDE are agroinfiltration experiments, which means that without a SP, CCP1 should accumulate in the cytoplasm and although this might explain why it can interact with cytoplasmic NL1-Pro, this is not the subcellular location of CCP1, given that it has its own SP and given the shown colocalisation with ER and vacuolar markers.

The reported data are consistent with a similar study on Nla of WYMV inhibiting RD21 in wheat, which has just been published in NatComm 14:7773. It would be appropriate to discuss the findings in the context of this article.

**Part III – Minor Issues: Editorial and Data Presentation Modifications**

Reviewer #1: Line 32: delete “only”, while the interaction of NIa-Pro is convincingly supported by three different protein-protein interaction methods, the initial screening for interaction with other viral proteins may not have allowed the detection of other interactions using the Y2H (nuclear-located) approach.

Line 173: change “were served” to “served”

Line 199: change “a both significant increasement” to “a significant increased for both”

Lines 361, 536, 538: change “230K and 234D” to “K230 and D234”. Please check entire text and figure legends to ensure that there are no other occurrences.

Line 395: change “system leaves” to “systemic leaves”

Line 426: change “increasement” to “increase”

Fig 4E legend: I appreciate that the His Abs show unspecific binding for the GST protein but it would be useful to add a note to that effect in the legend. Perhaps something like “please note that the His Abs did show unspecific binding for the GST and GST-NIa-Pro proteins”

Reviewer #2: 1. Abstract: arms races

2. Fig. 1B, as explained in the responses by authors, CCP1 exists multiple forms. To avoid misunderstanding by readers in the future, authors should indicate which form of CCP1 supposed to be in the gel, although they may be hard to distinguish because of tiny band size differences. Alternatively, authors can write more clearly in lines 151-156. In the figure legends, authors should also add this information. Please check throughout the manuscript.

3. Regarding Fig.6A, strictly, potyvirus has two ORFs, a large polyprotein and PIPO. The large polyprotein is further processed into 10 mature proteins by own proteases, and PIPO is expressed as fusion with the N-terminal of P3 as P3N-PIPO (Vijayapalani et al. 2012, PLoS Pathogens). If authors want to show the 11 proteins encoded by SCMV, then P3N-PIPO is more appropriate, but not PIPO.

4. S4 Fig, please indicate which CMV gene in RNA 1/2/3 has been used for viral accumulation quantification by RT-qPCR.

5. S7 Fig, please indicate each band clearly. We can clearly see GUS-3Myc control show band in the FLAG-IP anti-myc gel, suggesting relatively weak interaction signal or this is a unspecific band?

6. S12 Fig, please indicate N- and C- termini, and more importantly, the mutation residues.

7. S18 Fig is not in order. It appears firstly in the discussion, and should be placed in order. The results shown in S18 Fig is interesting, raising another pathway that CCP1 can suppress SCMV infection. Of course, this does not affect the main conclusion in this MS.

Reviewer #3: (No Response)

PLOS authors have the option to publish the peer review history of their article (what does this mean?). If published, this will include your full peer review and any attached files.

Reviewer #1: **Yes: **Helene Sanfacon

Reviewer #2: No

Reviewer #3: No

Figure Files:

Data Requirements:

Reproducibility:

References:

---

## [Editor Report · Decision Letter 2]

1 Mar 2024

Dear Dr. ZHOU,

We are pleased to inform you that your manuscript 'NIa-Pro of sugarcane mosaic virus targets Corn Cysteine Protease 1 (CCP1) to undermine salicylic acid-mediated defense in maize' has been provisionally accepted for publication in PLOS Pathogens.

Best regards,

Sebastian Schornack, Ph.D.

Academic Editor

PLOS Pathogens

Sonja Best

Section Editor

PLOS Pathogens

Michael Malim

Editor-in-Chief

PLOS Pathogens

orcid.org/0000-0002-7699-2064
---

## [Editor Report · Acceptance letter]

10 Mar 2024

Dear Dr. ZHOU,

We are delighted to inform you that your manuscript, "NIa-Pro of sugarcane mosaic virus targets Corn Cysteine Protease 1 (CCP1) to undermine salicylic acid-mediated defense in maize," has been formally accepted for publication in PLOS Pathogens.

Best regards,

Michael Malim

Editor-in-Chief

PLOS Pathogens

orcid.org/0000-0002-7699-2064